# Bilevel Optimization for Adversarial Learning Problems: Sharpness, Generation, and Beyond

**Risheng Liu[†], Zhu Liu[†], Weihao Mao[‡], Wei Yao[‡§], Jin Zhang[‡§¶*]**
[†]School of Software Technology, Dalian University of Technology
[‡]Mathematical Department, Southern University of Science and Technology
[§]National Center for Applied Mathematics Shenzhen
[¶]Detection Institute for Advanced Technology Longhua-Shenzhen (DIATLHSZ)
rsliu@dlut.edu.cn, liuzhu@mail.dlut.edu.cn, mwhaea123456@google.com,
yaow@sustech.edu.cn, zhangj9@sustech.edu.cn

## Abstract

Adversarial learning is a widely used paradigm in machine learning, often formulated as a min-max optimization problem where the inner maximization imposes adversarial constraints to guide the outer learner toward more robust solutions. This framework underlies methods such as Sharpness-Aware Minimization (SAM) and Generative Adversarial Networks (GANs). However, traditional gradient-based approaches to such problems often face challenges in balancing accuracy and efficiency due to second-order complexities. In this paper, we propose a bilevel optimization framework that reformulates these adversarial learning problems by leveraging the tractability of the lower-level problem. The bilevel framework introduces no additional complexity and enables the use of advanced bilevel tools. We further develop a provably convergent single-loop stochastic algorithm that effectively balances learning accuracy and computational cost. Extensive experiments show that our method improves generation quality of GANs, and consistently achieves higher accuracy for SAM under label noise and across various backbones, while promoting flatter loss landscapes. Overall, this work provides a practical and theoretically grounded framework for solving adversarial learning tasks through bilevel optimization.

## 1 Introduction

Nowadays, adversarial learning has attracted considerable attention due to its broad applicability in machine learning, particularly in addressing fairness and robustness. Typically formulated as a min-max optimization problem [46, 80, 16], adversarial learning involves maximizing an inner objective to impose adversarial constraints, thereby guiding the outer learner toward more robust solutions. This framework underpins a wide range of applications. In Generative Adversarial Networks (GANs) [8, 33, 41, 66, 77, 79], a generator (minimizer) and discriminator (maximizer) compete to improve the quality of synthetic data. In Sharpness-Aware Minimization (SAM) [2, 68, 69], adversarial perturbations on model parameters encourage optimization in flatter loss regions. In adversarial training [62, 46, 1, 80], input-space attacks are maximized to enhance model robustness against perturbations. These examples highlight the foundational role of adversarial learning across various domains.

Despite the widespread adoption and their empirical success, the traditional min-max framework introduces significant challenges for scalability, primarily because each training step involves

---

[*]Corresponding Author

multiple nested maximization steps. This results in prohibitively high computational costs, rendering these methods impractical for many real-world applications and further understandings [32, 80, 61, 64, 82].

Among the various applications of adversarial learning, we use Sharpness-Aware Minimization (SAM) as a representative example to highlight the challenges inherent in the min-max formulation and to motivate the need for a new reformulation.

**Sharpness-Aware Minimization.** Due to the influence of noise and overparameterization, modern neural networks are often prone to overfitting, resulting in poor generalization performance [78, 58, 68]. This challenge remains a significant barrier to further progress in machine learning. Recent numerical and theoretical studies have revealed a strong correlation between a model's generalization ability and the sharpness of the loss function landscape around its optimized parameters [28, 12, 14, 50, 22]. Classical optimization methods, which aim solely to minimize the value of the loss function, often fail to account for this geometric aspect and are thus insufficient for improving generalization.

To overcome this, researchers have turned their attention to the geometry of the loss surface, particularly the sharpness near minimizers, leading to the formulation of a min-max problem. This direction has culminated in the development of *Sharpness-Aware Minimization* (SAM), which explicitly incorporates sharpness into the optimization process [71, 83, 16, 51]. Recent work by [59] further suggests that SAMs effectiveness lies in its ability to balance the quality of diverse features, rather than merely seeking flatter minima.

**Difficulties in SAM: Min-Max Perspective.** Despite the rapid development and empirical success of SAM, algorithms based on SAMs original min-max formulation require implicit differentiation, which is computationally demanding. To reduce cost, practical implementations often avoid this step, but at the expense of accuracy, even with various refinements [16, 13, 5, 68]. For example, [72] incorporated second-order information to improve efficiency. Most theoretical analyses, however, still rely on the full SAM formulation without simplifications [5, 68, 69, 11]. For instance, [16] derived a PAC-Bayes bound for an averaged direction model, while [68] showed SAMs local regularization effect on the Hessian eigenvalues. These results point to a gap between simplified practical algorithms and the theoretical models that justify them.

**Difficulties in Adversarial Learning: the Min-Max Perspective.** The challenges encountered in SAM are not isolated but reflect a broader set of issues faced in many adversarial learning tasks. Similar difficulties appear in fast adversarial training [80, 1], generative adversarial learning [7, 64, 61, 82]. These methods frequently exhibit the following limitations [32, 80]: (i) reduced computational cost at the expense of learning accuracy, and (ii) a lack of sufficient theoretical guarantees.

These recurring issues raise a fundamental question:

*Is there an efficient framework that balances computational cost and learning accuracy in such adversarial learning tasks?*

To address this, we propose a unified bilevel optimization perspective, where the lower-level problem is chosen to be simple and, in many cases, admits a closed-form solution. This design allows us to apply advanced bilevel optimization theory and algorithms without increasing computational burden, as the lower-level remains analytically tractable. As a result, the proposed framework offers a principled and efficient approach applicable to various tasks, including SAM and generative adversarial networks, as discussed in this paper.

**Contributions.** We summarize our contributions as follows:

**Formulation-wise.** We propose a unified bilevel optimization framework to reformulate a range of adversarial learning tasks, including SAM and GANs, by introducing an efficient lower-level problem. This formulation enables the use of advanced bilevel optimization algorithms, achieving a better balance between computational efficiency and learning accuracy.

**Experiment-wise.** We conduct comprehensive experiments to evaluate the proposed bilevel framework, focusing on Sharpness-Aware Minimization (SAM) and generative adversarial learning (GAN). The results demonstrate that our approach effectively balances computational cost and numerical accuracy, highlighting its practicality and robustness in real-world adversarial learning tasks.

## 2 Related work

**Examples from the Traditional Perspective.** In recent years, mathematical formulations and corresponding optimization algorithms have played an increasingly prominent role in adversarial learning. Prominent examples of min-max formulations include adversarial training [78, 70, 1] and generative adversarial learning [55, 24, 49, 27, 26, 48, 25, 9, 23, 76]. While these approaches often reduce computational complexity, they tend to compromise model accuracy and robustness.

As an illustrate example, [16] leveraged the relationship between flat minima and generalization error to train deep neural networks (DNNs) with improved generalization across natural distributions using SAM. Similarly, [71] proposed a method that regularizes the sharpness term in adversarial training, achieving significantly more robust generalization against adversarial attacks. A substantial body of research has explored combining SAM with other training strategies or neural architectures [10, 67, 63]. For instance, [31] improved SAM by scaling the sharpness adjustment relative to the parameter size, while [42] optimized computational efficiency by reusing previously computed weight perturbations. Numerous efforts have been made to refine and adapt the SAM algorithm. For example, [13] introduced the Efficient Sharpness-Aware Minimizer (ESAM), which incorporates two key training strategies: Stochastic Weight Perturbation (SWP) and Sharpness-Sensitive Data Selection (SDS). [52] provides a unified framework of analysis on SAM.

**Bilevel Optimization.** Bilevel optimization is a hierarchical framework involving an upper-level and a lower-level problem. It is a highly challenging yet impactful area in both theory and practice. This framework has led to significant progress in machine learning domains such as hyperparameter selection [53, 17, 45] and meta-learning [6, 30, 56, 17, 36, 21, 75, 81]. [74] proposed a bilevel algorithm for the 0-1 classification problem, and [80] introduced a bilevel reformulation of fast adversarial training originally modeled as a traditional min-max problem in [46]. In this work, we take the step toward establishing a bilevel perspective for adversarial learning tasks, and propose corresponding algorithms and theoretical analyses to overcome the limitations of traditional perspectives.

## 3 The Bilevel Perspective Beyond the Traditional Framework

In this section, we first revisit the traditional optimization approach commonly used in adversarial learning tasks, using Sharpness-Aware Minimization (SAM) as a representative example, and point out their key limitations. We then introduce a bilevel optimization perspective featuring an efficiently solvable lower-level problem. This reformulation enables the application of advanced bilevel methods without incurring additional computational complexity, offering a principled and practical framework for a broad class of adversarial learning problems.

### 3.1 Preliminary on The Traditional Perspective: Min-Max Formulations

We begin by introducing a widely used min-max optimization formulation that underlies several adversarial learning applications, such as adversarial training [78, 70, 1], generative adversarial learning [55, 24, 49, 27, 26, 48, 25, 9, 23, 76], and, as discussed in the next subsection, SAM.

Consider the standard min-max formulation for many adversarial learning tasks:

$$\min_{\omega} \max_{\delta \in \mathcal{C}} L(\omega, \delta), \tag{1}$$

where the parameter $\omega \in \mathbb{R}^n$, $\delta \in \mathcal{C} \subset \mathbb{R}^m$, $\mathcal{C}$ represents the regular closed constraint set. Let $\mathcal{S} = \bigcup_{i=1}^{N} \{(u_i, v_i)\}$ represents the training dataset sampled from the data space $\mathcal{U} \times \mathcal{V}$. The function $L : \mathbb{R}^n \times \mathbb{R}^m \to \mathbb{R}$ is the empirical loss defined as

$$L(\omega, \delta) = \frac{1}{N} \sum_{i=1}^{N} L_S(\omega, \delta, u_i, v_i),$$

where $L_S : \mathbb{R}^n \times \mathbb{R}^m \times \mathcal{U} \times \mathcal{V} \to \mathbb{R}_+$ is the per-sample loss. The goal is to learn optimal parameters $\omega$ and $\delta$ that solve the $\min$-$\max$ objective (1).

The loss function $L(\omega, \delta)$ is often nonconvex with respect to both $\omega$ and $\delta$. This nonconvexity implies that local or global minimizers with identical loss values may exhibit vastly different neighborhood behaviors and theoretical capabilities.

An example of the computational challenges in $\min$-$\max$ problems is provided by [73], where 128 GPUs were used to run Adversarial Training (AT) on the ImageNet dataset. Such computational requirements highlight the prohibitive costs of solving large-scale $\min$-$\max$ problems directly. Several computational techniques have been proposed to improve the efficiency of $\min$-$\max$ optimization. For instance, gradient alignment (GA) regularization [1] and fast adversarial training methods [70] have been developed to address these challenges. However, these methods often suffer from practical issues such as instability, catastrophic overfitting, and degraded robustness [32, 80]. These limitations underscore the need for more robust and scalable approaches, such as the bilevel perspective.

## 3.2 SAM: An Example of The Traditional Perspective

In this subsection, we present an example of Sharpness-Aware Minimization (SAM) [16], which is based on the observation that the generalization ability of a model is closely related to the sharpness of its loss function landscape.

Consider a local neighborhood around a parameter $\omega$ with radius $r > 0$ and a unit direction $\delta$. SAM introduces the following min-max formulation:

$$\min_{\omega} \left[ L(\omega) + \max_{\|\delta\| \leq 1} [L(\omega + r\delta) - L(\omega)] \right], \tag{2}$$

which can be equivalently written as:

$$\min_{\omega} \max_{\|\delta\| \leq 1} L(\omega + r\delta), \tag{3}$$

where $\mathcal{C} = \{\delta \in \mathbb{R}^n : \|\delta\| \leq 1\}$ is the set of feasible perturbations, and $r$ denotes the radius. This formulation augments the original loss function $L(\omega)$ by incorporating a sharpness term $L(\omega + r\delta) - L(\omega)$, which captures the worst-case increase in the loss within the neighborhood defined by $r\delta$. The goal of SAM is thus twofold: minimize the loss function while simultaneously reducing sharpness in the worst-case direction.

The common strategy to address these computational challenges is to adopt approximation techniques. In the context of (2), the maximum function $L_{\max} : \mathbb{R}^n \to \mathbb{R}$ is defined as:

$$L_{\max}(\omega) := \max_{\|\delta\| \leq 1} L(\omega + r\delta), \tag{4}$$

where $r$ denotes the neighborhood radius. The traditional SAM algorithm minimizes $L_{\max}(\omega)$ by employing an approximation technique and a subsequent *discard* process.

**Numerical Technique**    The approximation begins with a first-order Taylor expansion around $\omega$:

$$L(\omega + r\delta) \approx L(\omega) + r\delta^\top \nabla_\omega L(\omega). \tag{5}$$

Using this approximation, the maximizer $\delta^*(\omega)$ is computed as:

$$\delta^*(\omega) = \arg \max_{\|\delta\| \leq 1} \left[ L(\omega) + r\delta^\top \nabla_\omega L(\omega) \right] = \frac{\nabla_\omega L(\omega)}{\|\nabla_\omega L(\omega)\|}.$$

Substituting $\delta^*(\omega)$ back into the SAM formulation, it becomes:

$$\min_{\omega} L_{\text{ASC}}(\omega) := L\left( \omega + r\frac{\nabla_\omega L(\omega)}{\|\nabla_\omega L(\omega)\|} \right). \tag{6}$$

To efficiently compute the gradient, traditional methods calculate:

$$\begin{aligned} \nabla_\omega L\big(\omega + r\delta^*(\omega)\big) &= \frac{d\big(\omega + r\delta^*(\omega)\big)}{d\omega} \nabla_\omega L\big(\omega + r\delta^*(\omega)\big) \\ &= \nabla_\omega L\big(\omega + r\delta^*(\omega)\big) + r\frac{d\delta^*(\omega)}{d\omega} \nabla_\omega L\big(\omega + r\delta^*(\omega)\big). \end{aligned} \tag{7}$$

The second term, $r\frac{d\delta^*(\omega)}{d\omega} \nabla_\omega L\big(\omega + r\delta^*(\omega)\big)$, involves the derivative of the composition function $\delta^*(\omega)$, which is computationally expensive to evaluate. Therefore, traditional methods often discard

this term to simplify the gradient computation [16, 68, 29]. Thus, the standard iterative process can be expressed as:

$$\omega^{k+1} = \omega^k - t\nabla L\left(\omega^k + r\frac{\nabla L(\omega^k)}{\|\nabla L(\omega^k)\|}\right), \tag{8}$$

where $t > 0$ is the step size. Convergence analysis for this iterative scheme and its extensions has been extensively studied (e.g., [16, 29, 52]).

**Importance of Hessian.** To better understand the behavior of SAM, consider Table 1. In the table, $\lambda_1$ denotes the largest eigenvalue, $\lambda_{\min}$ the smallest non-zero eigenvalue, Tr the trace operator, and $H_L(\omega)$ the Hessian of the loss function $L$ at $\omega$.

Table 1: Definitions and biases of different SAM loss formulations [68].

| Type of Loss | Definition | Biases |
|---|---|---|
| Worst | $\max\limits_{\|\delta\|\leq 1} L(\omega + r\delta)$ | $\min\limits_{\omega} \lambda_1(H_L(\omega))$ |
| Ascent | $L\left(\omega + r\frac{\nabla L(\omega)}{\|\nabla L(\omega)\|}\right)$ | $\min\limits_{\omega} \lambda_{\min}(H_L(\omega))$ |
| Average | $\mathbb{E}_{(u,v)\sim\mathcal{D}}[L_S(\omega, u, v)]$ | $\min\limits_{\omega} \mathrm{Tr}(H_L(\omega))$ |

In Table 1, the *worst-direction loss* captures the sharpest curvature of the loss landscape and is associated with $\lambda_1$. The *ascent-direction loss* reflects the local curvature along the most favorable direction of increase and corresponds to $\lambda_{\min}$. Finally, the *average-direction loss* characterizes the overall sharpness of the loss surface and is measured by $\mathrm{Tr}(H_L(\omega))$. From this table, it is clear that the effectiveness of the SAM technique in minimizing sharpness arises from its incorporation of second-order information via the Hessian matrix $H_L(\omega)$. Specifically, the largest eigenvalue $\lambda_1(H_L(\omega))$ directly measures the sharpness of the loss function. This connection explains why SAM improves generalization by reducing sharpness. In fact, prior studies (e.g., [44, 4, 43]) have used $\lambda_1(H_L(\omega))$ as a key metric to evaluate SAM's efficiency in enhancing generalization performance.

**Loss of Accuracy.** While the discarding step reduces computational costs, it introduces inaccuracies that may compromise learning performance. Specifically, in (7), we have

$$\frac{d\delta^*(\omega)}{d\omega} = \nabla_\omega\left(\frac{\nabla L(\omega)}{\|\nabla L(\omega)\|}\right) = \frac{1}{\|\nabla L(\omega)\|}\left(H_L(\omega) - \frac{\nabla L(\omega)\nabla L(\omega)^\top H_L(\omega)}{\|\nabla L(\omega)\|^2}\right), \tag{9}$$

where $H_L(\omega)$ denotes the Hessian of the loss function $L$ (details are in Subsection 3.1). The term discarded in (7) contains valuable geometric information about the loss landscape, which plays a critical role in establishing theoretical guarantees and ensuring robust optimization [68, 69]. This discrepancy between the theoretical formulation and its practical implementation highlights the need for a more principled and efficient framework. Experiments in Subsection 5.1 also shows that its loses accuracy.

### 3.3 The Bilevel Perspective

We introduce the bilevel perspective to address the limitations of traditional approaches, offering a simple lower-level problem without introducing additional complexities.

**Bilevel Perspective.** The bilevel optimization framework provides a natural and effective solution to the challenges in traditional perspectives. This formulation facilitates the application of advanced methods, such as those based on the Moreau envelope. Crucially, the lower-level problem remains simple and tractable, avoiding unnecessary complexity. This structure allows for the preservation of learning accuracy while managing computational costs, making it a robust alternative to traditional formulations.

Specifically, by selecting a suitable lower-level function $L_\ell : \mathbb{R}^n \times \mathbb{R}^m \to \mathbb{R}$, we reformulate traditional models - such as the min-max problem (1) – into a bilevel optimization problem:

$$\min_\omega L(\omega, \tilde{\delta})$$
$$\text{subject to } \tilde{\delta} = \arg\min_{\delta\in\mathcal{C}}\{L_\ell(\omega, \delta)\}, \tag{10}$$

where the lower-level problem allows efficient computation of the optimal solution $\tilde{\delta}$ for a given $\omega$, and $\mathcal{C}$ denotes the feasible set for $\delta$.

As an example, consider the SAM formulation in (2) and (6). By setting $L_\ell(\omega, \delta) := -\delta^T \nabla L(\omega)$, the lower-level problem becomes a linear program over the unit ball $\mathcal{C}$. This admits a closed-form solution: $\tilde{\delta} = \frac{\nabla L(\omega)}{\|\nabla L(\omega)\|}$. Hence, this bilevel reformulation introduces no additional complexity compared to the original model. In this case, the bilevel formulation (10) is equivalent to the traditional SAM model (2). The bilevel perspective allows both theoretical analysis and numerical optimization tools from bilevel programming to be applied effectively to such tasks.

## 4 Algorithm and Theoretical Investigation

Given the bilevel perspective, we now investigate algorithms tailored for this type of bilevel optimization. Solving large-scale bilevel optimization (BLO) problems for complex learning tasks presents two significant challenges: ensuring computational efficiency and guarantees of accuracy. To address these challenges, we apply a stochastic single-loop algorithm adopted from [40], inspired by [18], based on the Moreau envelope.

**Value Function.** One of the main challenges in solving bilevel programming problems (BLPPs) lies in their nested structure: one must first solve the lower-level problem and then optimize the upper-level objective based on that solution.

Let the value function be defined as

$$V(\omega) := \min_{\delta \in \mathcal{C}} L_\ell(\omega, \delta).$$

Then, the bilevel problem (10) can be equivalently reformulated as:

$$\begin{aligned} &\min_{\omega} \ L(\omega, \delta) \\ &\text{subject to} \quad L_\ell(\omega, \delta) - V(\omega) \leq 0, \quad \delta \in \mathcal{C}. \end{aligned} \tag{11}$$

However, the nonsmooth and nonconvex nature of $V(\omega)$ poses significant computational challenges.

**Moreau Envelope.** To address this issue, we reformulate the problem using the Moreau envelope of $L_\ell(\omega, \delta)$ with parameter $\gamma > 0$:

$$v_\gamma(\omega, \delta) := \inf_{\theta \in \mathcal{C}} \left\{ L_\ell(\omega, \theta) + \frac{1}{2\gamma} \|\theta - \delta\|^2 \right\}. \tag{12}$$

The Moreau envelope provides a smooth approximation to the value function. Geometrically, it defines a family of smooth surfaces that approximate the graph of $V$. Using this, we reformulate problem (10) as:

$$\begin{aligned} &\min_{\omega} \ L(\omega, \delta) \\ &\text{subject to} \quad L_\ell(\omega, \delta) - v_\gamma(\omega, \delta) \leq 0, \quad \delta \in \mathcal{C}. \end{aligned} \tag{13}$$

If $L_\ell(\omega, \cdot)$ is convex in $\delta$ for each fixed $\omega$, then (10) and (13) are equivalent [18, Theorem 2.1]. If $L_\ell(\omega, \cdot)$ is only weakly convex, then the latter is a relaxation of the former [40, Theorem A.1].

Note that $L_\ell(\omega, \delta) - v_\gamma(\omega, \delta) \geq 0$ for all $(\omega, \delta) \in \mathbb{R}^n \times \mathcal{C}$, so the constraint is effectively an equality constraint. This violates standard constraint qualifications and complicates the derivation of optimality conditions.

To circumvent this, we introduce a penalty parameter $\mu > 0$ and consider the penalized problem:

$$\min_{\omega} \ F_\mu(\omega, \delta) := L(\omega, \delta) + \mu \left[ L_\ell(\omega, \delta) - v_\gamma(\omega, \delta) \right], \quad \text{subject to} \quad \delta \in \mathcal{C}. \tag{14}$$

**Algorithm.** Based on the Moreau envelope, we propose a single-loop stochastic algorithm. At iteration $k$, given a mini-batch $b_k \subset \{1, \ldots, N\}$, define:

$$L^{b_k}(\omega, \delta) := \frac{1}{|b_k|} \sum_{i \in b_k} L_S(\omega, \delta, u_i, v_i), \tag{15}$$

where $L_S$ denotes the sample-wise loss. The batch-based lower-level loss is denoted as $L_\ell^{b_k}(\omega, \delta)$. The update is computed using gradients of $L^{b_k}(\omega, \delta)$, as described in (18).

Unlike SAM [17, 40] or fast adversarial training [80], our method avoids computing gradients of composite functions, thus significantly reducing computational cost while preserving robustness and accuracy.

**Convergence Analysis.** We define the residual function using the selected mini-batch $b_k$:

$$R_k^{b_k}(\omega, \delta) := \text{dist}\left(0, \nabla L^{b_k}(\omega, \delta) + \mu_k \left(\nabla L_\ell^{b_k}(\omega, \delta) - \nabla v_\gamma^{b_k}(\omega, \delta)\right) + N_{\mathbb{R}^n \times \mathcal{C}}(\omega, \delta)\right),$$

where $N_{\mathbb{R}^n \times \mathcal{C}}(\omega, \delta)$ is the normal cone to $\mathbb{R}^n \times \mathcal{C}$ at $(\omega, \delta)$.

When $b_k = \{1, \dots, N\}$, denote

$$R_k(\omega, \delta) := \text{dist}\left(0, \nabla L(\omega, \delta) + \mu_k \left(\nabla L_\ell(\omega, \delta) - \nabla v_\gamma(\omega, \delta)\right) + N_{\mathbb{R}^n \times \mathcal{C}}(\omega, \delta)\right).$$

This residual serves as a stationarity measure for the penalized problem:

$$\min_\omega F_{\mu_k}(\omega, \delta) := L(\omega, \delta) + \mu_k \left[L_\ell(\omega, \delta) - v_\gamma(\omega, \delta)\right], \quad \text{subject to} \quad \delta \in \mathcal{C}. \tag{16}$$

A point $(\omega, \delta)$ is stationary for (16) if and only if $R_k(\omega, \delta) = 0$.

**Special Case: SAM.** In the SAM setting where $L(\omega, \delta) = L(\omega + r\delta)$ and $L_\ell(\omega, \delta) = -\delta^\top \nabla_\omega L(\omega)$, the lower-level solution is given by:

$$\delta = \theta_\gamma^*(\omega, \delta) = \frac{\nabla L(\omega)}{\|\nabla L(\omega)\|}.$$

Then the gradient of the penalized objective becomes:

$$\nabla F_\mu(\omega, \delta) = \left(\nabla L(\omega + r\delta),\ r\nabla L(\omega + r\delta) - \mu \nabla L(\omega)\right).$$

Thus $R_k(\omega, \delta)$ captures the sharpness in the ascent direction model (6). We then present the following convergence result for the proposed algorithm:

**Theorem 4.1.** *Assume the algorithm satisfies some standard conditions. Then for any $p \in (0, \frac{1}{2})$, we have:*

$$\mathbb{E}\left[\min_{0 \le k \le K} R_k^{b_k}(\omega^{k+1}, \delta^{k+1})\right] = O\left(\frac{1}{K^{(1-2p)/2}}\right). \tag{17}$$

Figure 1 compares the convergence performance of our algorithm with existing bilevel methods, demonstrating its superior efficiency.

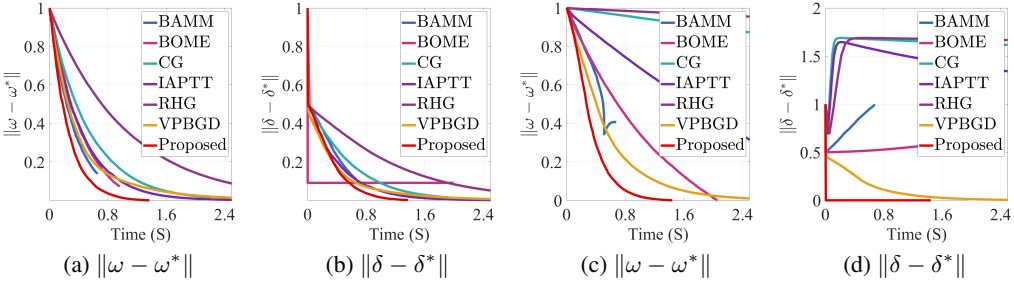

(a) $\|\omega - \omega^*\|$     (b) $\|\delta - \delta^*\|$     (c) $\|\omega - \omega^*\|$     (d) $\|\delta - \delta^*\|$

Figure 1: Illustration of convergence behaviors compared with representative BLO schemes.

**Future Generalization.** For various types of min-max problems, modern methods have been proposed (e.g., [1, 2, 32, 80, 65]). For instance, [65] investigates min-max problems of the form $\min_\theta \max_\phi L(F(\theta), G(\phi))$ in the convex-concave setting. Our method can be applied to some of these problems, particularly when the inner concave subproblem $\max_\phi L(F(\theta), G(\phi))$ is tractable, but solving the overall min-max problem requires sacrificing learning accuracy for computational efficiency. Some of these problems suffer from cyclic behavior of level sets (see Figure 4 in [65]). Unlike the approach in [65], we address this issue through a bilevel reformulation, whose convergence is established in Theorem A.8. In such cases, our bilevel framework provides a compelling alternative, offering a more favorable balance between learning accuracy and computational cost.

# 5 Experiments

In this section, we validate the performance under two real-world learning applications, including generative adversarial network and sharpness-ware minimization. Then we demonstrate the superiority of proposed scheme, illustrating the convergence behaviors and computation efficiency on synthetic numerical problems compared with existing BLO schemes. Details of implementation configurations and parameters selection are provided in the Appendix B. The source codes will be released at `https://github.com/LiuZhu-CV/BLOAL`.

Table 2: Evaluation of the robustness for SAM under varying noise labels and diverse backbones.

| Noise Label | SGD | SAM | Ours | Backbone | SGD | SAM | Ours |
|---|---|---|---|---|---|---|---|
| Clean | $95.47 \pm 0.12$ | $96.27 \pm 0.03$ | **96.34±0.11** | ResNet34 | $95.58 \pm 0.15$ | $96.61 \pm 0.08$ | **$97.75 \pm 0.06$** |
| 10% | $89.58 \pm 0.19$ | $92.84 \pm 0.55$ | **93.41±0.18** | ResNet50 | $95.40 \pm 0.44$ | $96.48 \pm 0.05$ | **$96.50 \pm 0.09$** |
| 20% | $82.64 \pm 0.31$ | $90.80 \pm 0.73$ | **91.48±0.06** | ResNet101 | $95.66 \pm 0.18$ | $96.60 \pm 0.13$ | **$96.69 \pm 0.05$** |

Table 3: Evaluation of the robustness based on sharpness metrics and diverse perturbation rates.

| Sharpness Metric | SGD | SAM | Ours | Perturbation | SAM | Ours |
|---|---|---|---|---|---|---|
| Hessian Norm | $8.869 \pm 3.31$ | $4.412 \pm 1.84$ | **3.840±1.63** | $r=0.05$ | $96.22 \pm 0.13$ | **$96.23 \pm 0.10$** |
| Trace | $13756 \pm 1710$ | $6650 \pm 1828$ | **5023±1101** | $r=0.15$ | $96.25 \pm 0.05$ | **$96.38 \pm 0.26$** |
| Top Eigenvalues | $1048.347$ | $621.349$ | **520.889** | $r=0.2$ | $96.26 \pm 0.05$ | **$96.28 \pm 0.17$** |

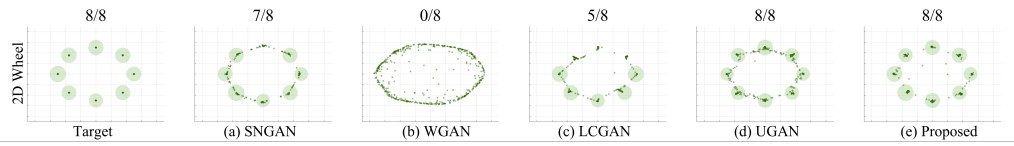

Figure 2: Comparison of generative adversarial learning under synthesized 2D wheels.

## 5.1 Real-world Applications

**Generative Adversarial Learning.** We conduct comparison with Stacked MNIST, a challenging dataset with 1000 modes and two-dimensional simulation experiments based on Gaussian distribution, generating eight distribution of 2D wheels. Table 4 reports the numerical performances compared with specialized methods, in terms of KL divergence and the maximum of modes. The visualization of data generation is shown in Figure 2. It can be observed that the proposed scheme can effectively generate all modes, compared with SNGAN [49], WGAN [3], LCGAN [15] and UGAN [47].

Table 4: Performance comparison of GAN on StackedMNIST.

| Methods | Modes | $D_{KL}$ |
|---|---|---|
| DcGAN [54] | 99 | 3.40 |
| VEEGAN [60] | 150 | 2.95 |
| WGAN [3] | 959 | 0.73 |
| PacGAN [34] | 992 | 0.28 |
| R3GAN [20] | 1000 | 0.12 |
| Ours | 1000 | 0.08 |

**Sharpness-ware Minimization.** As aforementioned, we propose a BLO perspective for SAM with computation accuracy and efficiency. Table 2 illustrates the robustness of different optimizer under varying levels of label noise and backbones on the Cifar-10 benchmark. Compared to standard SGD and SAM, our method consistently achieves superior accuracy across all noise labels and backbone, demonstrating its adaptability and robustness. Table 3 presents the analyses of model robustness based on sharpness-aware metrics, specifically analyzing the Hessian properties and the impact of varying perturbation rates $r$. The Hessian Norm quantifies the overall curvature of the loss landscape. The trace of the Hessian matrix reflects the sum of its eigenvalues. Lower top eigenvalues correspond to smoother loss surfaces. These metrics indicates our method encourages convergence toward flatter, more robust regions in the parameter space. Furthermore, Ours achieves consistently higher accuracy than SAM, demonstrating the practical robustness against diverse perturbation radius.

## 5.2 Synthetic Numerical Evaluation

**General Min-Max Numerical Cases.** Here, we compare the convergence behavior of our method with representative bilevel optimization (BLO) schemes, including Hessian-based algorithms (*e.g.,*

RHG [17], CG [53], IAPTT [37]), first-order methods (*e.g.*, BOME [35], VPBGD [57]), and the single-loop approach BAMM [39].

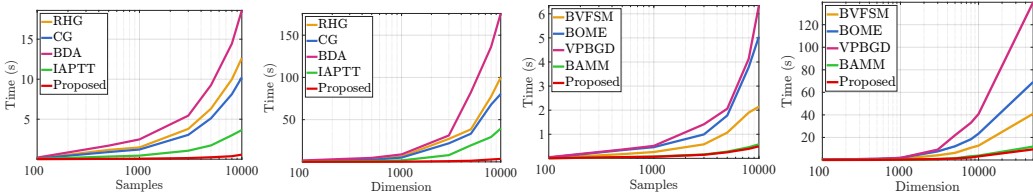

(a) Comparison with Hessian-based algorithms    (b) Comparison with Hessian-free algorithms

Figure 3: Computation efficiency comparison as the dimension of samples and features increase.

The first min-max numerical example is $\min_{\omega \in \mathbb{R}} \max_{\delta \in \mathbb{R}} \omega^2 - \delta^2 + \alpha \sin(\omega\delta)$, where $\alpha$ controls the nonlinearity. This problem features bilinear coupling and oscillatory components, resulting in multiple saddle points that can hinder convergence. For simplicity, we set $\alpha = 1$ and constrain $\omega, \delta \in [0, 1]$, with the optimal solution at $(0, 0)$. Figure 1 (a) and (b) show the convergence of various BLO methods in terms of $\|\omega - \omega^*\|$ and $\|\delta - \delta^*\|$. Our method achieves the fastest convergence across both metrics, while BOME and VPBGD exhibit slower decay and higher residuals. We also consider a more challenging case: $\min_{\omega} \max_{\delta} \omega^T A\delta + \sin(\|\omega\|^2 - \|\delta\|^2)$, where $\omega, \delta, A \in \mathbb{R}$. The nonlinear squared coupling introduces varying oscillation frequencies and degenerate critical points. With $\omega, \delta \in [0, 1]$ and $A = 1$, the solution remains $(0, 0)$. Figure 1 (c) and (d) illustrate that our method maintains superior convergence speed and stability, while other approaches, including BAMM, BOME, CG, and VPBGD, suffer from oscillations in the lower-level updates.

**Large-scale Computation Efficiency.** We construct a synthetic numerical case to perform the large-scale computation, which is formulated as $\min_{\omega} \max_{\delta} \sum_{i=1}^{m} \sum_{j=1}^{n} (\omega_{i,j}^2 - \delta_{i,j}^2 + \sin(\omega_{i,j}\delta_{i,j}))$, where $m$ and $n$ denote the numbers of samples and dimensions. Figure 3 illustrates the computation time of single step under varying numbers of samples and feature dimensions. Compared with Hessian-free algorithms, value function based schemes (*e.g.,* BVFSM [38], BOME and VPBGD) suffer from steep computational increases, especially with larger feature dimensions. Moreover, compared with single-loop scheme BAMM, the proposed method remains highly efficient, improving 16.8% at 10000 samples.

# 6    Conclusion

This paper presents a unified bilevel optimization framework for solving adversarial learning tasks, aiming to balance computational efficiency and learning accuracy. Motivated by the limitations of traditional methods, such as high cost and limited theoretical grounding, our approach offers an efficient and interpretable alternative. In particular, it effectively addresses Sharpness-Aware Minimization (SAM) and generative adversarial learning tasks. Extensive experiments show that our method captures all data modes in generative modeling and improves FID and JS scores. For SAM, it consistently achieves higher accuracy under label noise and across different backbones, while encouraging flatter loss landscapes. Overall, this work provides a practical and theoretically sound foundation for addressing adversarial learning tasks via bilevel optimization.

# Acknowledgment

This work is partially supported by the National Natural Science Foundation of China (Nos.62450072, U22B2052, 624B2033, 12371305, 12222106, 12326605), the Distinguished Youth Funds of the Liaoning Natural Science Foundation (No.2025JH6/101100001), the Distinguished Young Scholars Funds of Dalian (No.2024RJ002), the Fundamental Research Funds for the Central Universities, Guangdong Basic and Applied Basic Research Foundation (No.2022B1515020082) and the Longhua District Science and Innovation Commission Project Grants of Shenzhen (No.20250113G43468522). We thank the anonymous reviewers for their valuable comments and constructive suggestions on this work. Authors listed in alphabetical order.

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

# A Convergence Analysis

This part will present some classical and basic properties in the convergence analysis for this paper.

**Preliminaries for Convergence Analysis**

In this subsection, we present essential properties of the Moreau envelope and its gradient, which form the foundation for our convergence analysis. Detailed proofs of the following results can be found in [18, 40].

We now recall a regularity result showing that the Moreau envelope inherits convexity under weak convexity of the original function.

**Lemma A.1.** *In (13), suppose $L_\ell(\omega, \delta)$ is weakly convex in $\omega$ and $\delta$ with constants $\rho_1, \rho_2 \geq 0$, respectively, and $\gamma \in (0, \frac{1}{2\rho_2})$. Then, for any $\rho_{v_1} \geq \rho_1$ and $\rho_{v_2} \geq \frac{1}{\gamma}$, the function*

$$v_\gamma(\omega, \delta) + \frac{\rho_{v_1}}{2}\|\omega\|^2 + \frac{\rho_{v_2}}{2}\|\delta\|^2$$

*is convex over $\mathbb{R}^n \times \mathbb{R}^n$.*

Next, we state a key result on the differentiability of the Moreau envelope, which allows its gradient to be expressed via the minimizer of the inner problem.

**Lemma A.2.** *Under the assumptions of Lemma A.1, the solution of the minimization problem*

$$S_\gamma(\omega, \delta) := \arg\min_{\theta \in \mathcal{C}} \left\{ L_\ell(\omega, \theta) + \frac{1}{2\gamma}\|\theta - \delta\|^2 \right\}$$

*is unique for all $(\omega, \delta)$. Let $\theta_\gamma^*(\omega, \delta)$ denote this unique minimizer. Then the Moreau envelope $v_\gamma$ is differentiable and satisfies*

$$\nabla v_\gamma(\omega, \delta) = \left( \nabla_\omega L_\ell(\omega, \theta_\gamma^*(\omega, \delta)), \frac{1}{\gamma}(\delta - \theta_\gamma^*(\omega, \delta)) \right).$$

Finally, we establish a Lipschitz-type bound for the smoothed minimizer $\theta_\gamma^*(\omega, \delta)$, which is critical for bounding gradient approximation errors.

**Lemma A.3.** *Let $\gamma \in \left( 0, \frac{1}{2\rho_2} \right)$ and define $L_\theta := \max\{1, \gamma\ell\}$. Then for any $(\omega, \delta), (\bar{\omega}, \bar{\delta}) \in \mathbb{R}^n \times \mathbb{R}^m$, we have*

$$\|\theta_\gamma^*(\omega, \delta) - \theta_\gamma^*(\bar{\omega}, \bar{\delta})\| \leq L_\theta \left( \|\omega - \bar{\omega}\| + \|\delta - \bar{\delta}\| \right).$$

These results provide the theoretical basis for analyzing the approximation error and stability of our proposed algorithm.

**Gradient of the Normalized Gradient**

In this subsection, we calculate the gradient of the discarded term $\frac{d\delta^*(\omega)}{d\omega}$ in (7) in details. Recall that $\omega \in \mathbb{R}^n$ and $L : \mathbb{R}^n \to \mathbb{R}$ be a smooth function. Define the normalized gradient mapping as:

$$f(\omega) := \frac{\nabla L(\omega)}{\|\nabla L(\omega)\|}.$$

We aim to compute the Jacobian $\nabla_\omega f(\omega) \in \mathbb{R}^{n \times n}$. Denote:

$$g(\omega) := \nabla L(\omega), \quad h(\omega) := \|g(\omega)\| = \|\nabla L(\omega)\|.$$

Then we have:

$$f(\omega) = \frac{g(\omega)}{h(\omega)}.$$

Using the quotient rule for vector-valued functions:

$$\nabla_\omega \left( \frac{g(\omega)}{h(\omega)} \right) = \frac{1}{h(\omega)} \nabla_\omega g(\omega) - \frac{g(\omega)}{h(\omega)^2} \nabla_\omega h(\omega)^T.$$

We now compute the two components separately:

- The Jacobian of $g(\omega)$ is the Hessian matrix:

$$\nabla_\omega g(\omega) = \nabla^2 L(\omega) =: H_L(\omega).$$

- The gradient of the scalar function $h(\omega)$ is given by:

$$\nabla_\omega h(\omega) = \nabla_\omega \|g(\omega)\| = \frac{H_L(\omega)\nabla L(\omega)}{\|\nabla L(\omega)\|}.$$

Substituting into the quotient rule, we obtain:

$$\nabla_\omega \left( \frac{\nabla L(\omega)}{\|\nabla L(\omega)\|} \right) = \frac{1}{\|\nabla L(\omega)\|} H_L(\omega) - \frac{\nabla L(\omega)}{\|\nabla L(\omega)\|^2} \left( \frac{H_L(\omega)\nabla L(\omega)}{\|\nabla L(\omega)\|} \right)^T$$

$$= \frac{1}{\|\nabla L(\omega)\|} \left( H_L(\omega) - \frac{\nabla L(\omega)\nabla L(\omega)^\top H_L(\omega)}{\|\nabla L(\omega)\|^2} \right).$$

This expression gives the Jacobian matrix of the normalized gradient $f(\omega)$, which appears frequently in sharpness-aware optimization and directional smoothing.

### Algorithms

In this subsection, we underly our algorithm. The algorithm follows a standard stochastic process. Given batches $b_k \subset \{1, \ldots, N\}$ at each iteration, the procedure is summarized as follows:

- **Initialization:** Set step sizes $\{\alpha_k\}$, $\{\beta_k\}$, $\{\eta_k\}$, and penalty parameters $\{\mu_k\}$, and initialize the variables $\omega^0$, $\delta^0$, and $\theta^0$.
- **Update rules:**

$$\theta^{k+1} = \text{Proj}_{\mathcal{C}} \left( \theta^k - \eta_k [\nabla_\delta L_\ell^{b_k}(\omega^k, \theta^k) + \frac{1}{\gamma}(\theta^k - \delta^k)] \right),$$

$$\delta^{k+1} = \text{Proj}_{\mathcal{C}} \left( \delta^k - \beta_k \left[ \frac{1}{\mu_k} \nabla_\delta L^{b_k}(\omega^k, \delta^k) + \nabla_\delta L_\ell^{b_k}(\omega^k, \delta^k) - \frac{1}{\gamma}(\delta^k - \theta^{k+1}) \right] \right),$$

$$\omega^{k+1} = \omega^k - \alpha_k \left[ \frac{1}{\mu_k} \nabla_\omega L^{b_k}(\omega^k, \delta^{k+1}) + \nabla_\omega L_\ell^{b_k}(\omega^k, \delta^{k+1}) - \nabla_\omega L_\ell^{b_k}(\omega^k, \theta^{k+1}) \right].$$

$$(18)$$

### Algorithm for SAM

In particular, we consider the SAM problem given in (3). For a fixed pair $(\omega, \delta)$, the Moreau envelope reformulates the lower-level problem as the following smooth optimization problem:

$$\min_{\theta \in \mathcal{C}} L_\ell(\omega, \theta) + \frac{1}{2\gamma} \|\theta - \delta\|^2, \tag{19}$$

which is typically convex in $\theta$. In this case, any Karush-Kuhn-Tucker (KKT) point corresponds to a global minimizer. The global minimizer $\theta_\gamma^*(\omega, \delta)$ of (19) satisfies the optimality condition:

$$0 \in \nabla_\delta L_\ell(\omega, \theta^*) + \frac{1}{\gamma}(\theta^* - \delta) + N(\theta^*, \mathcal{C}),$$

where $\mathcal{C} = \{\delta \in \mathbb{R}^n : \|\delta\| \leq 1\}$, and $N(\theta^*, \mathcal{C})$ denotes the normal cone to $\mathcal{C}$ at $\theta^*$, defined by:

$$N(\theta^*, \mathcal{C}) = \begin{cases} \{0\}, & \|\theta^*\| < 1; \\ \{\lambda\theta^* : \lambda \geq 0\}, & \|\theta^*\| = 1. \end{cases} \tag{20}$$

Consequently, the closed-form expression for the global minimizer $\theta_\gamma^*(\omega, \delta)$ is:

$$\theta_\gamma^*(\omega, \delta) = \begin{cases} \delta + \gamma\nabla L(\omega), & \|\delta + \gamma\nabla L(\omega)\| < 1; \\ \frac{\delta + \gamma\nabla L(\omega)}{\|\delta + \gamma\nabla L(\omega)\|}, & \|\delta + \gamma\nabla L(\omega)\| \geq 1. \end{cases} \tag{21}$$

Then we adopt the following algorithm:

- **Initialization:** Set step sizes $\{\alpha_k\}$, $\{\beta_k\}$, $\{\eta_k\}$, and penalty parameters $\{\mu_k\}$, and initialize the variables $\omega^0$, $\delta^0$, and $\theta^0$.

- **Update:**

$$\theta^{k+1} = \begin{cases} \delta^k + \gamma \nabla L^{b_k}(\omega^k), & \|\delta^k + \gamma \nabla L^{b_k}(\omega^k)\| < 1, \\ \frac{\delta^k + \gamma \nabla L^{b_k}(\omega^k)}{\|\delta^k + \gamma \nabla L^{b_k}(\omega^k)\|}, & \|\delta^k + \gamma \nabla L^{b_k}(\omega^k)\| \geq 1. \end{cases}$$

- **Generate i.i.d. Gaussian vectors:** Generate a sequence of independent and identically distributed (i.i.d.) Gaussian vectors $\{u_{k,j} \in \mathbb{R}^n\}_{j=1}^Q$, where each $u_{k,j}$ is sampled from a standard normal distribution. Update the parameters as follows:

$$\delta^{k+1} = \frac{\nabla L^{b_k}(\omega^k)}{\|\nabla L^{b_k}(\omega^k)\|},$$

$$\omega^{k+1} = \omega^k - \alpha_k \left[ \frac{1}{\mu_k} \nabla_\omega L^{b_k}(\omega^k + r\delta^{k+1}) + H_{L^{b_k}}(\omega^k)(\theta^{k+1} - \delta^{k+1}) \right]$$

$$\approx \omega^k - \alpha_k \left[ \frac{1}{\mu_k} \nabla_\omega L^{b_k}(\omega^k + r\delta^{k+1}) + J^*(\omega^k, \theta^{k+1} - \delta^{k+1}) \right]$$

$$= \omega^k - \alpha_k \left[ \frac{1}{\mu_k} \nabla_\omega L^{b_k}(\omega^k + r\delta^{k+1}) \right.$$

$$\left. + \frac{1}{Q} \sum_{j=1}^Q \left\langle \frac{\nabla L^{b_k}(\omega^k + \mu u_{k,j}) - \nabla L^{b_k}(\omega^k)}{\mu}, \theta^{k+1} - \delta^{k+1} \right\rangle u_{k,j} \right].$$

Here, $H_L$ denotes the Hessian matrix, and $J^*$ is an approximation matrix used for numerical purposes, as described in [19].

## Convergence Result with Biased Estimator

In this section, we present the convergence analysis of the proposed algorithm. For convergence analysis, we impose the following assumptions on the bilevel formulation (10):

**Assumption A.4.** The lower-level function $L_\ell(\omega, \delta)$ is locally Lipschitz continuous with constant $\ell$.

**Assumption A.5.** The functions $L(\omega, \delta)$ and $L_\ell(\omega, \delta)$ in (10) are weakly convex with respect to both $\omega$ and $\delta$, with constants $\rho_{L,1}, \rho_{L,2}, \rho_1, \rho_2 \geq 0$, respectively. That is, the functions

$$L(\omega, \delta) + \rho_{L,1}\|\omega\|^2 + \rho_{L,2}\|\delta\|^2 \quad \text{and} \quad L_\ell(\omega, \delta) + \rho_1\|\omega\|^2 + \rho_2\|\delta\|^2$$

are convex.

The analysis relies on the following assumption regarding the variance of the biased gradient estimators.

**Assumption A.6** (Biased Estimator). For any batch $b \subset \{1, \dots, N\}$, the following bounds on the expected squared bias hold for $\sigma_\omega, \sigma_\delta, \sigma_{\ell,\omega}, \sigma_{\ell,\delta} \geq 0$:

$$\begin{aligned} \mathbb{E}\left[\|\nabla_\omega L^b(\omega, \delta) - \nabla_\omega L(\omega, \delta)\|^2\right] &\leq \sigma_\omega^2, \\ \mathbb{E}\left[\|\nabla_\delta L^b(\omega, \delta) - \nabla_\delta L(\omega, \delta)\|^2\right] &\leq \sigma_\delta^2, \\ \mathbb{E}\left[\|\nabla_\omega L_\ell^b(\omega, \delta) - \nabla_\omega L_\ell(\omega, \delta)\|^2\right] &\leq \sigma_{\ell,\omega}^2, \\ \mathbb{E}\left[\|\nabla_\delta L_\ell^b(\omega, \delta) - \nabla_\delta L_\ell(\omega, \delta)\|^2\right] &\leq \sigma_{\ell,\delta}^2. \end{aligned} \tag{22}$$

Given a penalty parameter $\mu > 0$, we define:

$$f_\mu(\omega, \delta) := \frac{1}{\mu} L(\omega, \delta) + L_\ell(\omega, \delta) - v_\gamma(\omega, \delta),$$

$$F_\mu(\omega, \delta) := L(\omega, \delta) + \mu[L_\ell(\omega, \delta) - v_\gamma(\omega, \delta)].$$

The convergence analysis will be conducted primarily based on these two functions. Set

$$d_\delta^k := \frac{1}{\mu_k}\nabla_\delta L^{b_k}(\omega^k, \delta^k) + \nabla_\delta L_\ell^{b_k}(\omega^k, \delta^k) - \frac{1}{\gamma}(\delta^k - \theta^{k+1});$$

$$d_\omega^k := \frac{1}{\mu_k}\nabla_\omega L^{b_k}(\omega^k, \delta^{k+1}) + \nabla_\omega L_\ell^{b_k}(\omega^k, \delta^{k+1}) - \nabla_\omega L_\ell^{b_k}(\omega^k, \theta^{k+1}). \tag{23}$$

The following lemma shows that the expected value of $f_{\mu_k}$ exhibits a monotonic decreasing behavior, which plays a critical role in the convergence analysis.

**Lemma A.7.** *Under Assumptions A.4, A.5, A.6 and assumptions in Lemma A.2, the sequence* $(\omega^k, \delta^k)$ *generated by the algorithm satisfies*

$$\mathbb{E}[f_{\mu_k}(\omega^{k+1}, \delta^{k+1}) - f_{\mu_k}(\omega^k, \delta^k)] \leq \left[\frac{L_{f_k}}{2} - \frac{1}{8\alpha_k}\right]\|\omega^k - \omega^{k+1}\|^2$$

$$+ \left[\frac{1}{2}L_{f_k} - \frac{1}{4\beta_k} + \frac{\alpha_k \ell^2 L_\theta^2}{2}\right]\|\delta^k - \delta^{k+1}\|^2$$

$$+ \left(\frac{\alpha_k \ell^2}{2} + \frac{\beta_k}{2\gamma^2}\right)\|\theta^{k+1} - \theta_\gamma^*(\omega^k, \delta^k)\|^2 \tag{24}$$

$$+ \frac{2\alpha_k}{\mu_k^2}\sigma_\omega^2 + 4\alpha_k\sigma_{\ell,\omega}^2 + \frac{2\beta_k}{\mu_k^2}\sigma_\delta^2 + 2\beta_k\sigma_{\ell,\delta}^2.$$

*where $L_{f_k}$ is the Lipschitz constant of the gradient of $f_{\mu_k}$ in its second-order Taylor expansion.*

*Proof.* Assuming that $f_{\mu_k}$ is smooth, we have the inequality with $L_{f_k} > 0$:

$$f_{\mu_k}(\omega^{k+1}, \delta^{k+1}) \leq f_{\mu_k}(\omega^k, \delta^{k+1}) + \langle\nabla_\omega f_{\mu_k}(\omega^k, \delta^{k+1}), \omega^{k+1} - \omega^k\rangle + \frac{L_{f_k}}{2}\|\omega^{k+1} - \omega^k\|^2$$

$$= f_{\mu_k}(\omega^k, \delta^{k+1}) + \langle\nabla_\omega f_{\mu_k}(\omega^k, \delta^{k+1}) - d_\omega^k + d_\omega^k, \omega^{k+1} - \omega^k\rangle + \frac{L_{f_k}}{2}\|\omega^{k+1} - \omega^k\|^2$$

$$= f_{\mu_k}(\omega^k, \delta^{k+1}) + \langle\nabla_\omega f_{\mu_k}(\omega^k, \delta^{k+1}) - d_\omega^k, \omega^{k+1} - \omega^k\rangle + \left(\frac{L_{f_k}}{2} - \frac{1}{\alpha_k}\right)\|\omega^{k+1} - \omega^k\|^2. \tag{25}$$

By Assumption A.6, the expectation of the first inner product term satisfies:

$$E\left[\langle\nabla_\omega f_{\mu_k}(\omega^k, \delta^{k+1}) - d_\omega^k, \omega^{k+1} - \omega^k\rangle\right]$$

$$\leq E[\langle\frac{1}{\mu_k}(\nabla_\omega L(\omega^k, \delta^{k+1}) - \nabla_\omega L^{b_k}(\omega^k, \delta^{k+1})) + (\nabla_\omega L_\ell(\omega^k, \delta^{k+1}) - \nabla_\omega L_\ell^{b_k}(\omega^k, \delta^{k+1}))$$

$$- (\nabla_\omega L_\ell(\omega^k, \theta_\gamma^*(\omega^k, \delta^{k+1})) - \nabla_\omega L_\ell^{b_k}(\omega^k, \theta^{k+1})), \omega^{k+1} - \omega^k\rangle]$$

$$\leq \frac{2\alpha_k}{\mu_k^2}\sigma_\omega^2 + \frac{1}{8\alpha_k}\|\omega^{k+1} - \omega^k\|^2 + 2\alpha_k\sigma_{\ell,\omega}^2 + \frac{1}{8\alpha_k}\|\omega^{k+1} - \omega^k\|^2$$

$$+ 2\alpha_k\sigma_{\ell,\omega}^2 + \frac{1}{8\alpha_k}\|\omega^{k+1} - \omega^k\|^2 + \ell\|\theta^{k+1} - \theta_\gamma^*(\omega^k, \delta^{k+1})\|\|\omega^{k+1} - \omega^k\|$$

$$\leq \frac{7}{8\alpha_k}\|\omega^{k+1} - \omega^k\|^2 + \frac{\alpha_k\ell^2}{2}\|\theta^{k+1} - \theta_\gamma^*(\omega^k, \delta^{k+1})\|^2 + \frac{2\alpha_k}{\mu_k^2}\sigma_\omega^2 + 4\alpha_k\sigma_{\ell,\omega}^2. \tag{26}$$

Similarly, we apply the same approach for the update of $\delta^{k+1}$ and obtain:

$$f_{\mu_k}(\omega^k, \delta^{k+1}) \leq f_{\mu_k}(\omega^k, \delta^k) + \langle\nabla_\delta f_{\mu_k}(\omega^k, \delta^k), \delta^{k+1} - \delta^k\rangle + \frac{L_{f_k}}{2}\|\delta^{k+1} - \delta^k\|^2$$

$$= f_{\mu_k}(\omega^k, \delta^k) + \langle\nabla_\delta f_{\mu_k}(\omega^k, \delta^k) - d_\delta^k + d_\delta^k, \delta^{k+1} - \delta^k\rangle + \frac{L_{f_k}}{2}\|\delta^{k+1} - \delta^k\|^2 \tag{27}$$

$$= f_{\mu_k}(\omega^k, \delta^k) + \langle\nabla_\delta f_{\mu_k}(\omega^k, \delta^k) - d_\delta^k, \delta^{k+1} - \delta^k\rangle + \left(\frac{L_{f_k}}{2} - \frac{1}{\beta_k}\right)\|\delta^{k+1} - \delta^k\|^2,$$

and

$$E\left[\langle \nabla_\delta f_{\mu_k}(\omega^k, \delta^k) - d_\delta^k, \delta^{k+1} - \delta^k \rangle\right]$$

$$\leq \frac{2\beta_k}{\mu_k^2}\sigma_\delta^2 + \frac{1}{8\beta_k}\|\delta^{k+1} - \delta^k\|^2 + 2\beta_k\sigma_{\ell,\delta}^2 + \frac{1}{8\beta_k}\|\delta^{k+1} - \delta^k\|^2 + \frac{1}{2\beta_k}\|\delta^{k+1} - \delta^k\|^2$$

$$+ \frac{\beta_k}{2\gamma^2}\|\theta^{k+1} - \theta_\gamma^*(\omega^k, \delta^k)\|^2 \tag{28}$$

$$\leq \frac{2\beta_k}{\mu_k^2}\sigma_\delta^2 + 2\beta_k\sigma_{\ell,\delta}^2 + \frac{3}{4\beta_k}\|\delta^{k+1} - \delta^k\|^2 + \frac{\beta_k}{2\gamma^2}\|\theta^{k+1} - \theta_\gamma^*(\omega^k, \delta^k)\|^2.$$

Combining all parts, we obtain the claimed result. $\qquad\square$

Using Lemma A.7, we obtain the following convergence result for the proposed algorithm:

**Theorem A.8.** *Suppose $\gamma \in (0, \frac{1}{2\rho_2})$ and penalty parameters satisfy $\bar{\mu}(k+1)^p \geq \mu_{k+1} \geq \mu_k \geq \underline{\mu} > 0$. Then, for any $0 < \underline{\alpha} < \alpha_k < \frac{1}{8L_{f_k}}$, $0 < \underline{\beta} < \beta_k < \frac{1}{4L_{f_k}}$, and $0 < \eta_k < \frac{1}{\rho_{L,2}+2\rho_2}$ the iterates $\{(\omega^k, \delta^k)\}$ generated by the algorithm satisfy*

$$\mathbb{E}\left[\frac{1}{K^{1/2}}\min_{0 \leq k \leq K} R_k^{b_k}(\omega^{k+1}, \delta^{k+1})\right] = O\left(\frac{1}{K^{1/2}}\right). \tag{29}$$

*Furthermore, if the step sizes satisfy $\sum_{k=0}^\infty (\alpha_k + \beta_k) < \infty$, then for any $p \in (0, \frac{1}{2})$, it holds that*

$$\mathbb{E}\left[\min_{0 \leq k \leq K} R_k^{b_k}(\omega^{k+1}, \delta^{k+1})\right] = O\left(\frac{1}{K^{(1-2p)/2}}\right). \tag{30}$$

*Proof.* By [18, Proposition 2.1,],[40, Lemma A.7], we deduce that $(e_\omega^k, e_\delta^k) \in \nabla R_k^{b_k}(\omega^{k+1}, \delta^{k+1})$ with

$$e_\omega^k = \nabla_\omega F_{\mu_k}^{b_k}(\omega^{k+1}, \delta^{k+1}) - \mu_k d_\omega^k - \frac{\mu_k}{\alpha_k}(\omega^{k+1} - \omega^k)$$

$$e_\delta^k = \nabla_\delta F_{\mu_k}^{b_k}(\omega^{k+1}, \delta^{k+1}) - \mu_k d_\delta^k - \frac{\mu_k}{\beta_k}(\delta^{k+1} - \delta^k). \tag{31}$$

Then by Lipschitz continuity of $F_{\mu_k}$ and $L_\ell$ with constants $L_{F_k}, \ell$, we have

$$\|e_\omega^k\| \leq \|\nabla_\omega F_{\mu_k}^{b_k}(\omega^{k+1}, \delta^{k+1}) - \nabla_\omega F_{\mu_k}^{b_k}(\omega^k, \delta^{k+1})\| + \|\nabla_\omega F_{\mu_k}^{b_k}(\omega^k, \delta^{k+1}) - \mu_k d_\omega^k\|$$

$$+ \|\frac{\mu_k}{\alpha_k}(\omega^{k+1} - \omega^k)\| \tag{32}$$

$$\leq \mu_k L_{F_k}\|\omega^{k+1} - \omega^k\| + \frac{\mu_k}{\alpha_k}\|\omega^{k+1} - \omega^k\| + \mu_k \ell\|\theta^{k+1} - \theta_\gamma^*(\omega^k, \delta^{k+1})\|,$$

$$\|e_\delta^k\| \leq \|\nabla_\delta F_{\mu_k}^{b_k}(\omega^{k+1}, \delta^{k+1}) - \nabla_\delta F_{\mu_k}^{b_k}(\omega^k, \delta^k)\| + \|\nabla_\delta F_{\mu_k}^{b_k}(\omega^k, \delta^k) - \mu_k d_\delta^k\|$$

$$+ \|\frac{\mu_k}{\beta_k}(\delta^{k+1} - \delta^k)\| \tag{33}$$

$$\leq \mu_k L_{F_k}\|(\omega^{k+1} - \omega^k, \delta^{k+1} - \delta^k)\| + \frac{\mu_k}{\beta_k}\|\delta^{k+1} - \delta^k\|$$

$$+ \frac{\mu_k}{\gamma}(\|\theta^{k+1} - \theta_\gamma^*(\omega^k, \delta^k)\| + L_\theta\|\omega^{k+1} - \omega^k\|).$$

Thus,

$$R_k^{b_k}(\omega^{k+1}, \delta^{k+1}) \leq \mu_k(2L_{F_k} + \frac{1}{\alpha_k} + \frac{L_\theta}{\gamma})\|\omega^{k+1} - \omega^k\| + \mu_k(L_{F_k} + \frac{1}{\beta_k})\|\delta^{k+1} - \delta^k\|$$

$$+ \mu_k(\ell\|\theta^{k+1} - \theta^*(\omega^k, \delta^{k+1})\| + \frac{1}{\gamma}\|\theta^{k+1} - \theta^*(\omega^k, \delta^k)\|). \tag{34}$$

By [40][Lemma A.6], there exists $\sigma_k \in (0, 1)$ such that

$$\|\theta^{k+1} - \theta_\gamma^*(\omega^k, \delta^k)\| \leq \sigma_k\|\theta^k - \theta_\gamma^*(\omega^k, \delta^k)\|.$$

Then we have

$$R_k^{b_k}(\omega^{k+1}, \delta^{k+1})$$

$$\leq \mu_k(2L_{F_k} + \frac{1}{\alpha_k} + \frac{L_\theta}{\gamma})\|\omega^{k+1} - \omega^k\| + \mu_k(L_{F_k} + \frac{1}{\beta_k} + L_\theta \ell)\|\delta^{k+1} - \delta^k\|$$

$$+ \mu_k(\ell + \frac{1}{\gamma})\|\theta^{k+1} - \theta_\gamma^*(\omega^k, \delta^k)\| \tag{35}$$

$$\leq \mu_k(2L_{F_k} + \frac{1}{\alpha_k} + \frac{L_\theta}{\gamma})\|\omega^{k+1} - \omega^k\| + \mu_k(L_{F_k} + \frac{1}{\beta_k} + L_\theta \ell)\|\delta^{k+1} - \delta^k\|$$

$$+ \mu_k\sigma_k(\ell + \frac{1}{\gamma})\|\theta^k - \theta_\gamma^*(\omega^k, \delta^k)\|.$$

Because $\alpha_k > \underline{\alpha} > 0, \beta_k > \underline{\beta} > 0$, there exists $C_R > 0, C_\theta > 0$ such that

$$\frac{1}{\mu_k^2}[R_k^{b_k}(\omega^{k+1}, \delta^{k+1})]^2$$

$$\leq C_R(\frac{1}{8\alpha_k}\|\omega^{k+1} - \omega^k\|^2 + \frac{1}{16\beta_k}\|\delta^{k+1} - \delta^k\|^2 + C_\theta\|\theta^k - \theta_\gamma^*(\omega^k, \delta^k)\|^2). \tag{36}$$

Since $\mu_k \leq \mu_{k+1}$, then

$$f_{\mu_{k+1}}(\omega^{k+1}, \delta^{k+1}) - f_{\mu_k}(\omega^{k+1}, \delta^{k+1}) = (\frac{1}{\mu_{k+1}} - \frac{1}{\mu_k})L(\omega^{k+1}, \delta^{k+1}) < 0. \tag{37}$$

By inserting (37) into Lemma A.7, we know

$$\mathbb{E}[f_{\mu_{k+1}}(\omega^{k+1}, \delta^{k+1}) + (\ell^2 + \frac{1}{\gamma^2})\|\theta^{k+1} - \theta_\gamma^*(\omega^{k+1}, \delta^{k+1})\|]$$

$$- \mathbb{E}[f_{\mu_k}(\omega^k, \delta^k) + (\ell^2 + \frac{1}{\gamma^2})\|\theta^k - \theta_\gamma^*(\omega^k, \delta^k)\|]$$

$$\leq \mathbb{E}[f_{\mu_k}(\omega^{k+1}, \delta^{k+1}) - f_{\mu_k}(\omega^k, \delta^k)] + (\ell^2 + \frac{1}{\gamma^2})[\|\theta^{k+1} - \theta_\gamma^*(\omega^{k+1}, \delta^{k+1})\|$$

$$- \|\theta^k - \theta_\gamma^*(\omega^k, \delta^k)\|] \tag{38}$$

$$\leq \left[\frac{L_{f_k}}{2} - \frac{1}{8\alpha_k}\right]\|\omega^k - \omega^{k+1}\|^2 + \left[\frac{1}{2}L_{f_k} - \frac{1}{4\beta_k} + \frac{\alpha_k\ell^2 L_\theta^2}{2}\right]\|\delta^k - \delta^{k+1}\|^2$$

$$+ (\frac{\alpha_k\ell^2}{2} + \frac{\beta_k}{2\gamma^2})\|\theta^{k+1} - \theta_\gamma^*(\omega^k, \delta^k)\|^2 + (\ell^2 + \frac{1}{\gamma^2})[\|\theta^{k+1} - \theta_\gamma^*(\omega^{k+1}, \delta^{k+1})\|^2$$

$$- \|\theta^k - \theta_\gamma^*(\omega^k, \delta^k)\|^2] + \frac{2\alpha_k}{\mu_k^2}\sigma_\omega^2 + 4\alpha_k\sigma_{\ell,\omega}^2 + \frac{2\beta_k}{\mu_k^2}\sigma_\delta^2 + 2\beta_k\sigma_{\ell,\delta}^2.$$

Besides, recall that by [40][Lemma A.6], there exists $\sigma_k \in (0, 1)$ such that

$$\|\theta^{k+1} - \theta_\gamma^*(\omega^k, \delta^k)\| \leq \sigma_k\|\theta^k - \theta_\gamma^*(\omega^k, \delta^k)\|.$$

Then for any $\epsilon_k > 0$, we have

$$\frac{1}{2}\alpha_k\|\theta^{k+1} - \theta_\gamma^*(\omega^k, \delta^k)\|^2 + \|\theta^{k+1} - \theta_\gamma^*(\omega^{k+1}, \delta^{k+1})\|^2 - \|\theta^k - \theta_\gamma^*(\omega^k, \delta^k)\|^2$$

$$\leq (\epsilon_k + \frac{1}{2}\alpha_k)\|\theta^{k+1} - \theta_\gamma^*(\omega^k, \delta^k)\|^2 - \|\theta^k - \theta_\gamma^*(\omega^k, \delta^k)\|^2$$

$$+ \frac{1}{\epsilon_k}\|\theta_\gamma^*(\omega^{k+1}, \delta^{k+1}) - \theta_\gamma^*(\omega^k, \delta^k)\|^2 \tag{39}$$

$$\leq [\sigma_k^2(\epsilon_k + \frac{1}{2}\alpha_k) - 1]\|\theta^k - \theta_\gamma^*(\omega^k, \delta^k)\|^2 + \frac{L_\theta^2}{\epsilon_k}(\|\omega^{k+1} - \omega^k\|^2 + \|\delta^{k+1} - \delta^k\|^2)$$

Similarly, we have

$$
\begin{aligned}
&\frac{1}{2}\beta_k\|\theta^{k+1} - \theta_\gamma^*(\omega^k, \delta^k)\|^2 + \|\theta^{k+1} - \theta_\gamma^*(\omega^{k+1}, \delta^{k+1})\|^2 - \|\theta^k - \theta_\gamma^*(\omega^k, \delta^k)\|^2 \\
&\leq (\epsilon_k + \frac{1}{2}\beta_k)\|\theta^{k+1} - \theta_\gamma^*(\omega^k, \delta^k)\|^2 - \|\theta^k - \theta_\gamma^*(\omega^k, \delta^k)\|^2 \\
&\quad + \frac{1}{\epsilon_k}\|\theta_\gamma^*(\omega^{k+1}, \delta^{k+1}) - \theta_\gamma^*(\omega^k, \delta^k)\|^2 \\
&\leq [\sigma_k^2(\epsilon_k + \frac{1}{2}\beta_k) - 1]\|\theta^k - \theta_\gamma^*(\omega^k, \delta^k)\|^2 + \frac{L_\theta^2}{\epsilon_k}(\|\omega^{k+1} - \omega^k\|^2 + \|\delta^{k+1} - \delta^k\|^2)
\end{aligned}
\tag{40}
$$

Thus, when $\frac{1}{2}L_{f_k} + \frac{L_\theta^2}{\epsilon_k} \leq \frac{1}{16\alpha_k}$ and $\frac{1}{2}L_{f_k} + \frac{L_\theta^2}{\epsilon_k} \leq \frac{1}{8\beta_k}$ (by $\alpha_k < \frac{1}{8L_{f_k}}, \beta_k < \frac{1}{4L_{f_k}}$), we have

$$
\begin{aligned}
&\mathbb{E}[f_{\mu_{k+1}}(\omega^{k+1}, \delta^{k+1}) + (\ell^2 + \frac{1}{\gamma^2})\|\theta^{k+1} - \theta_\gamma^*(\omega^{k+1}, \delta^{k+1})\|\|] \\
&\quad - \mathbb{E}[f_{\mu_k}(\omega^k, \delta^k) + (\ell^2 + \frac{1}{\gamma^2})\|\theta^k - \theta_\gamma^*(\omega^k, \delta^k)\|\|] \\
&\leq \ell^2[\sigma_k^2(\epsilon_k + \frac{1}{2}\alpha_k) + \frac{1}{\gamma^2}\sigma_k^2(\epsilon_k + \frac{1}{2}\beta_k) - 1]\|\theta^k - \theta_\gamma^*(\omega^k, \delta^k)\|^2 \\
&\quad + [\frac{1}{2}L_{f_k} - \frac{1}{8\alpha_k} + \frac{L_\theta^2}{\epsilon_k}]\|\omega^{k+1} - \omega^k\|^2 + [\frac{1}{2}L_{f_k} - \frac{1}{4\beta_k} + \frac{L_\theta^2}{\epsilon_k}]\|\delta^{k+1} - \delta^k\|^2 \\
&\quad + \frac{2\alpha_k}{\mu_k^2}\sigma_\omega^2 + 4\alpha_k\sigma_{\ell,\omega}^2 + \frac{2\beta_k}{\mu_k^2}\sigma_\delta^2 + 2\beta_k\sigma_{\ell,\delta}^2 \\
&\leq -C_R(C_\theta\|\theta^k - \theta_\gamma^*(\omega^k, \delta^k)\|^2 + \frac{1}{16\alpha_k}\|\omega^{k+1} - \omega^k\|^2 + \frac{1}{8\beta_k}\|\delta^{k+1} - \delta^k\|^2) \\
&\quad + \frac{2\alpha_k}{\mu_k^2}\sigma_\omega^2 + 4\alpha_k\sigma_{\ell,\omega}^2 + \frac{2\beta_k}{\mu_k^2}\sigma_\delta^2 + 2\beta_k\sigma_{\ell,\delta}^2.
\end{aligned}
\tag{41}
$$

As a result, by take the sum of iteratives $k$ in (36), when $\max\{\sigma_\omega^2, \sigma_\delta^2, \sigma_{\ell,\omega}^2, \sigma_{\ell,\delta}^2\} \leq \sigma^2, \mu_k \geq \underline{\mu} > 0$, there exists a constant $C > 0$ such that

$$
\begin{aligned}
&C\sum_{k=0}^K \frac{1}{\mu_k^2}\mathbb{E}[R_k^{b_k}(\omega^{k+1}, \delta^{k+1})]^2 \\
&\leq f_{\mu_0}(\omega^0, \delta^0) + (\ell^2 + \frac{1}{\gamma^2})\|\theta^0 - \theta_\gamma^*(\omega^0, \delta^0)\| + \sum_{k=0}^K[\frac{2\alpha_k}{\mu_k^2}\sigma_\omega^2 + 4\alpha_k\sigma_{\ell,\omega}^2 + \frac{2\beta_k}{\mu_k^2}\sigma_\delta^2 + 2\beta_k\sigma_{\ell,\delta}^2] \\
&\leq f_{\mu_0}(\omega^0, \delta^0) + (\ell^2 + \frac{1}{\gamma^2})\|\theta^0 - \theta_\gamma^*(\omega^0, \delta^0)\| + 2\sigma^2(\frac{1}{\underline{\mu}^2} + 1)\sum_{k=0}^K(2\alpha_k + \beta_k).
\end{aligned}
\tag{42}
$$

Because $\alpha_k < \frac{1}{8L_{f_k}}$ and $\beta_k < \frac{1}{4L_{f_k}}$, we have

$$
\begin{aligned}
&\frac{C}{K}\sum_{k=0}^K \frac{1}{\mu_k^2}\mathbb{E}[R_k^{b_k}(\omega^{k+1}, \delta^{k+1})]^2 \\
&\leq \frac{1}{K}[f_{\mu_0}(\omega^0, \delta^0) + (\ell^2 + \frac{1}{\gamma^2})\|\theta^0 - \theta_\gamma^*(\omega^0, \delta^0)\|\|] + 2\sigma^2(\frac{1}{\underline{\mu}^2} + 1)(\frac{1}{2L_{f_k}} + \frac{1}{4L_{f_k}}).
\end{aligned}
\tag{43}
$$

Then

$$
\frac{1}{K^{1/2}}\mathbb{E}[R_k^{b_k}(\omega^{k+1}, \delta^{k+1})] = O(\frac{1}{K^{1/2}}).
\tag{44}
$$

Furthermore, when $\sum_{k=0}^\infty (\alpha_k + \beta_k) < \infty$, we have

$$
\sum_{k=0}^\infty \frac{1}{\mu_k^2}\mathbb{E}[R_k^{b_k}(\omega^{k+1}, \delta^{k+1})]^2 < \infty.
\tag{45}
$$

For $p \in (0, \frac{1}{2})$, because $\mu_k \leq \bar{\mu}(k+1)^p$

$$\sum_{k=0}^{K} \frac{1}{\mu_k^2} \geq \frac{1}{\bar{\mu}^2} \sum_{k=0}^{K} (\frac{1}{k+1})^{2p} \geq \frac{1}{\bar{\mu}^2} \int_{1}^{K+2} \frac{1}{t^{2p}} dt \geq \frac{(K+2)^{1-2p} - 1}{(1-2p)\bar{\mu}^2}. \tag{46}$$

then

$$\mathbb{E}[\min_{0 \leq k \leq K} R_k^{b_k}(\omega^{k+1}, \delta^{k+1})] = O(\frac{1}{K^{(1-2p)/2}}). \tag{47}$$

$\square$

## B  Experimental Details

We conducted the experiments on a PC with Intel i5-13600KF CPU (3.5 GHz), 32GB RAM and NVIDIA RTX 4090 GPU. We leveraged the PyTorch framework on the 64-bit Linux system. In the following, we elaborately introduce the implementation details and parameter configurations.

**Numerical Cases**

As for the first case, we set $\eta$, $\beta$, $\alpha$, $\gamma$, $\mu$, and $p$ as 0.001, 0.01, 0.0001, 20, and 0.1 and leverage $\|\omega - \omega^*\| \leq 1e^{-4}$ as the stop criterion. SGD optimizer is used for the update of $\omega$. We set the the maximum steps of optimization as 1000 uniformly. As for the second case, $\eta$, $\beta$, $\alpha$, $\gamma$, $\mu$, and $p$ are 0.001, 0.05, 0.0001, 20, and 0.1, respectively.

**Generative Adversarial Learning**

The goal of generative adversarial learning is to build a min-max game between the generator $\text{Gen}(\omega; \cdot)$ and discriminator $\text{Dis}(\delta; \cdot)$, which can be formulated as $\min_{\omega} \max_{\delta} \log(\text{Dis}(\delta; \mathbf{u})) + \log(1 - \text{Dis}(\delta; \text{Gen}(\omega; \mathbf{v})))$, where $\mathbf{u} \sim p_{\text{data}}$ represents the real data distribution and $\mathbf{v}$ denotes the random latent vector. The hyperparameters $\eta$, $\beta$, $\alpha$, $\gamma$, $\mu$, and $p$ are set to 0.005, 0.005, 0.01, 100, 5 and 0.1, respectively. The generator is a three-layer fully connected neural network that maps an input vector to a target output through two hidden layers. Each hidden layer is followed by a non-linear activation function. The final layer outputs the generated vector without activation. The discriminator consists of two hidden layers and a final classification layer. All hidden layers use the ReLU activation function. The output is passed through a sigmoid activation. We conduct the experiments under 8 distribution of 2D wheel, sampling 500 data points from 2D surfaces. All experiments are repeated three times with diverse random seeds.

**Sharpness-aware Minimization**

We conduct image classification experiments using the standard open-source CIFAR-10 benchmark, which consists of 50,000 training and 10,000 testing image-label pairs. ResNet-18 is employed as the backbone model to evaluate performance under noisy labels and various perturbation radius $r$. For fair comparison in experiments involving both noisy labels and different backbone architectures, we fix $r = 0.1$. The hyperparameters $\alpha$, $\gamma$, $\mu$, $Q$, and $p$ are set to 0.05, $1 \times 10^{-4}$, 0.75, 1 and 0.01, respectively. Following the setup in [2], we apply basic augmentation during training, including horizontal flipping, four-pixel padding, and cropping. Models are trained from scratch for 200 epochs using a batch size of 128 and a cosine learning rate schedule. To ensure fair comparison, all experiments are repeated three times with diverse random seeds.

