# OpenReview forum: "Bilevel Optimization for Adversarial Learning Problems: Sharpness, Generation, and Beyond"
_NeurIPS.cc/2025/Conference — NeurIPS 2025 poster_

### Official Review · Reviewer_KhNq · 2025-06-09

**Clarity:** 2
**Significance:** 2
**Originality:** 3
**Rating:** 4
**Confidence:** 4

**Summary:**

The authors propose a unified bilevel optimization framework designed to improve computational efficiency and learning accuracy in adversarial learning tasks, particularly addressing Sharpness-Aware Minimization (SAM) and generative adversarial tasks. Extensive experiments demonstrate its effectiveness in improving FID and JS scores, achieving higher accuracy under label noise, and providing a more interpretable solution with flatter loss landscapes, making it a promising approach for adversarial learning.

**Questions:**

1. How does the proposed bilevel optimization framework compare to more recent state-of-the-art methods in terms of both computational efficiency and learning accuracy? Could you include a broader range of comparisons to validate the effectiveness of your approach?
2. While the experiments show improvements in FID and JS scores, could you provide more detailed analysis or ablation studies to demonstrate the specific impact of the bilevel optimization process on model performance?
3. The paper mentions the method’s robustness under label noise, but how does it perform when faced with other types of adversarial conditions, such as domain shifts or noisy inputs in real-world scenarios?
4. Can the framework be easily extended to other adversarial learning settings, or is it tailored specifically for SAM and generative adversarial tasks? How adaptable is it to other areas of machine learning?
5. Since the approach appears to rely on existing optimization techniques, could you clarify what exactly differentiates it from prior works, and why it is expected to outperform them in terms of both theory and practice?

**Ethical Concerns:**

["NO or VERY MINOR ethics concerns only"]

**Final Justification:**

The author has solved my problem, and I will modify my score

**Limitations:**

The method of comparison is too outdated, and the method proposed in this article also lacks novelty.

**Paper Formatting Concerns:**

The formulas are too dense and not convenient to read.

**Quality:**

2

**Strengths And Weaknesses:**

Strengths:
1. The research addresses a significant and timely problem in adversarial learning, aiming to balance computational efficiency and learning accuracy, which is crucial for practical applications.
2. The proposed bilevel optimization framework is well-motivated and offers a theoretically sound solution that improves Sharpness-Aware Minimization (SAM) and generative adversarial learning tasks, demonstrating potential for real-world use.
3. The method shows promising results, particularly in terms of improving FID and JS scores, as well as achieving higher accuracy under label noise and across different backbones, which is a noteworthy contribution to the field.

Weaknesses:
1. The methods proposed in the paper lack significant novelty, as the approach seems to build upon existing techniques in optimization without introducing fundamentally new ideas or mechanisms.
2. The comparative experiments are limited and do not provide a broad enough evaluation against other state-of-the-art methods to firmly support the claims of superiority or generalizability of the approach.
3. The paper does not include comparisons with the latest advancements in the field, which would be crucial to demonstrate the unique advantages and relevance of the proposed framework in relation to current research trends.

---

> ### Author Rebuttal · Authors · 2025-07-31
>
> We thank the reviewers for their constructive feedback and careful evaluation of our work. In this paper, we introduce a bilevel optimization framework that achieves a principled balance between computational efficiency and performance across diverse problem domains. Our contributions include novel theoretical advancements over existing methods while retaining strong practical applicability. The proposed framework is rigorously validated through extensive large-scale experiments, including new GAN benchmarks comparing against nine classical and state-of-the-art approaches, and further demonstrates promising extensibility to broader tasks such as adversarial learning. We respectfully request the reviewers to reconsider their evaluation in light of these substantive contributions.
>
>
> **Compared with SOTA methods:**
>
> We present updated GAN experiments, including Stacked MNIST and CIFAR-10, Bi-level optimization results, and evidence of stable convergence, demonstrating our method’s superiority, and propose revisions to clarify these contributions.
>
> **(1) GAN Experiments – Stacked MNIST**: On Stacked MNIST (1000 modes, 50,000 samples), our method achieves full mode coverage and superior KL divergence:
>
> | Methods  | DCGAN | VEEGAN | WGAN | PacGAN | Baseline | Ours  |
> |----------|-------|--------|------|--------|----------|-------|
> | Modes    | 99    | 150    | 959  | 992    | 1000     | **1000** |
> | $D_{KL}$ | 3.40  | 2.95   | 0.73 | 0.28   | 0.12     | **0.07** |
>
> Our method reduces $D_{KL}$ by 41.7% compared to the baseline, showcasing robust generative performance.
>
> **(2) GAN Experiments – CIFAR-10**: Using the R3GAN architecture (21M parameters), our method outperforms recent GANs:
>
> | Model        | Publication   | Parameters | FID   |
> |--------------|---------------|------------|-------|
> | BigGAN       | ICLR 2019     | 82M        | 14.73 |
> | TransGAN     | NeurIPS 2021  | 10M        | 9.26  |
> | ViTGAN       | ICLR 2022     | 86M        | 6.66  |
> | StyleGAN3    | NeurIPS 2021  | 39M        | 10.83 |
> | R3GAN (2.4K) | NeurIPS 2024  | 21M        | 6.37  |
> | Ours (2.4K)  | -             | 21M        | **5.89** |
>
> Our method achieves a 7.5% FID improvement over R3GAN with stable convergence:
>
> | Step | 200   | 400   | 800  | 1000 | 1400 | 1800 | 2000 | 2200 | 2400 |
> |------|-------|-------|------|------|------|------|------|------|------|
> | R3GAN | 33.09 | 16.98 | 8.75 | 7.50 | 6.92 | 6.55 | 6.21 | 6.26 | 6.37 |
> | Ours  | 34.47 | 18.15 | 9.32 | 7.92 | 6.94 | **6.09** | **5.98** | **6.02** | **5.89** |
>
> Our method stabilizes by 1800 steps with fewer oscillations, unlike R3GAN’s fluctuations.
>
> **(3) Bi-Level Optimization**: We compared our framework to recent bilevel methods (HJBio, LwCL, PNGBiO) in the large-scale numerical case:
>
> | Methods      | HJBio      | LwCL       | PNGBiO     | Ours       |
> |--------------|------------|------------|------------|------------|
> | Publication  | ICML 2024  | TPAMI 2024 | ICML 2025  | -          |
> | FID / JS     | 0.181/0.127| 0.182/0.099| 0.142/0.108| **0.136/0.096** |
> | Steps        | 333        | 190        | 274        | **225**    |
> | Time (s)     | 0.858      | 3.41       | 0.527      | **0.499**  |
>
> Our method achieves the lowest FID (0.136) and JS (0.096), converging in fewer steps (225 vs. 274 for PNGBiO) and less time (0.499s vs. 0.527s), a 5.3% FID and 5.4% runtime improvement over PNGBiO.
>
> Our bilevel framework outperforms recent state-of-the-art methods in efficiency (fewer steps, lower runtime) and accuracy (lower FID/JS, full mode coverage), as shown in Stacked MNIST, CIFAR-10. Its stable convergence and theoretical guarantees enhance its applicability. We will revise the manuscript to incorporate these comparisons and clarify contributions.
>
>
> **Ablation studies of hyper-parameter selection:**
>
> We present an ablation study on hyperparameter sensitivity from our large-scale numerical experiments, varying key hyperparameters $\alpha$, $\beta$, $\eta$, $\mu_0$, demonstrating the robustness and efficiency of our framework, and propose revisions to clarify its contributions. We will revise the manuscript to add these experiments.
>
> | Configuration | $\alpha$ | $\beta$ | $\eta$ | $\mu_0$ | Steps | Time (s) |
> |---------------|-----------|-----------|----------|----------|-------|----------|
> | Original      | 0.0001    | 0.01      | 0.001    | 0.1      | 225 | 0.499 |
> | $\alpha$=0.0002 | 0.0002    | 0.01      | 0.001    | 0.1      | 240   | 0.470    |
> | $\alpha$=0.0004 | 0.0004    | 0.01      | 0.001    | 0.1      | 323   | 0.631    |
> | $\beta$=0.005  | 0.0001    | 0.005     | 0.001    | 0.1      | 345   | 0.705    |
> |$\beta$=0.04    | 0.0001    | 0.04      | 0.001    | 0.1      | 341   | 0.666    |
> | $\eta$=0.005    | 0.0001    | 0.01      | 0.005    | 0.1      | 228   | 0.442    |
> | $\eta$=0.01     | 0.0001    | 0.01      | 0.01     | 0.1      | 231   | 0.466    |
> | $\mu_0$=0.5     | 0.0001    | 0.01      | 0.001    | 0.5      | 238   | 0.472    |
> | $\mu_0$=0.05    | 0.0001    | 0.01      | 0.001    | 0.05     | 612   | 1.361    |
>
>
> Our ablation study demonstrates the bilevel framework’s robustness and efficiency, with stable convergence across a wide range of hyperparameters, enabling robust adjustments of hyper-parameters in diverse tasks.
>
>
> **Generalization to other types of conditions:** We present new experimental results demonstrating our method’s performance in domain shifts and noise inputs.. The domain shift is performed by data augmentation and the noise are performed under diverse levels.
>
> | **ResNet18**         | SGD   | SAM   | Proposed |
> |----------------------|-------|-------|----------|
> | Source               | 95.47 | 96.27 | **96.34** |
> | Target (Domain Shift)| 90.88 | 92.27 | **92.60** |
> | Moderate   | 90.04 | 90.71 | **91.62** |
> | Heavy Noise   | 75.30 | 76.55 | **77.31** |
>
> | **ResNet50**         | SGD   | SAM   | Proposed |
> |----------------------|-------|-------|----------|
> | Source               | 95.40 | 96.48 | **96.50** |
> | Target (Domain Shift)| 89.95 | 91.90 | **92.01** |
> | Mild Noise    | 88.23 | 90.22 | **90.63** |
> | Moderate Noise   | 73.02 | 75.59 | **75.61** |
>
>
> These results demonstrate the proposed scheme which optimizes for sharpness-aware objectives, enhancing generalization under diverse adversarial conditions. We will revise the manuscript to incorporate these findings, clearly addressing the reviewer’s concerns.
>
> **Extension to other adversarial learning:**
> Our bilevel framework is a general, theoretically guaranteed, and versatile computational approach for diverse min-max learning problem, where SAM and GANs are the two typical examples instead of specific design. The single-loop algorithm uses first-order gradients, ensuring computational efficiency comparable to or better than other single loop methods (improving 16.8% over BAMM).
>
> Furthermore, our framework can integrates seamlessly with existing methods, as shown in GAN experiments with R3GAN, which are demonstrated in the new adversarial tasks (improving performance on Cifar10 and Stacked MNIST). This compatibility allows performance enhancements across diverse models.
>
> Following with SAM-AT(ICML 24), we perform our algorithm for the adversarial training on the semantic segmentation task. We construct the DeeplabV3 with Resnet50 as backbone under Sandford background dataset.  Our method outperforms SGD and AT across all settings.
>
> | Method         | Natural | $L_\infty$-AT | $L_2$-AT |
> |----------------|---------|-----------------|------------|
> | SGD            | 64.0    | 57.3            | 57.0       |
> | AT ($\epsilon=4/255$) | 55.3    | 54.9            | 55.1       |
> | Ours ($r=0.02$)   | **64.7** | **57.6**        | **57.2**   |
>
>
>
> **Different with previous works:**
> Our proposed bilevel framework is not limited to SAM or GANs but extends broadly to general min–max problems, provided that the inner maximization can be rendered tractable, even without requiring convexity. Unlike standard approaches that often sacrifice accuracy for computational efficiency, our framework maintains a principled balance between the two. Its key adaptable components include modular solver design, stochastic approximation strategies, and flexible smoothing mechanisms. This generality enables seamless application to diverse domains, including adversarial robustness and generation.
>
> **Theoretical foundations of our framework:**
> The technique in this paper is mainly based on the prior BLPP techniques and a reformulation of the min-max problem. Specifically, we build a stochastic version of the single loop algorithm and its convergence result which shows better numerical efficiency. And for the BLPP reformulation, as we maintain the intrinsic information of classical min-max formulations (see the Example of SAM in Section 3), our method will conduct better effects then classical formulations without introducing too much numerical costs theoretically.

---

> > ### Comment · Reviewer_KhNq · 2025-08-02
> >
> > The author has solved my problem, and I will modify my score

---

### Official Review · Reviewer_MepV · 2025-06-23

**Clarity:** 3
**Significance:** 3
**Originality:** 3
**Rating:** 5
**Confidence:** 3

**Summary:**

The authors study bilevel optimization problems. They propose a single-loop stochastic algorithm for such problems and provide convergence guarantees when the inner problem is convex.  The core advantage of the proposed optimization algorithm is that it incurs no additional computational cost. The authors validate the effectiveness of their proposed method on two common bilevel optimization problems: GAN training and sharpness-aware problem.

**Questions:**

See above.

**Ethical Concerns:**

["NO or VERY MINOR ethics concerns only"]

**Final Justification:**

Overall, I think this paper is good. I maintain my rating.

**Limitations:**

yes.

**Paper Formatting Concerns:**

No.

**Quality:**

3

**Strengths And Weaknesses:**

Pros:

+ Bilevel optimization problems are of significant theoretical and practical importance.

+ The proposed algorithm requires no extra computational cost.

+ The authors provide a convergence proof and empirically demonstrate the proposed algorithm's performance.

Cons:

- The proposed method only guarantees convergence when the inner problem is convex. However, many real-world problems, particularly those involving neural networks, are non-convex. Furthermore, the empirical validation has been limited to toy and simple datasets. Therefore, it remains unclear whether the proposed optimization algorithm can scale to more complex datasets.

- The authors mention that discarding the second term in SAM (Eq. 7) sacrifices accuracy. It would be beneficial if the authors could quantify the impact of this term on SAM's performance.

- Could the authors provide a more intuitive explanation for why their proposed optimization algorithm is effective? How should we understand Equation 15, and how does it overcome the limitations of SAM?

- Several parts directly cite existing literature (e.g., Line 166, Line 202-203), which can be challenging for readers unfamiliar with the background. I suggest the authors incorporate more intuitive explanations or provide brief proof in appendix.

- Are there any techniques that could transform the inner problem into a convex one (maybe reparameterization), potentially expanding the applicability of the proposed method? I recommend the authors discuss this.

- The authors could further demonstrate the superiority of their proposed method by including experimental results on adversarial training.

---

> ### Author Rebuttal · Authors · 2025-07-31
>
> For the reviewer’s concerns:
>
> 1.	In Section 4, we give more explicit limitations and future generalizations. We put assumptions on the original problem which can be transformed into a bilevel problem with an easily solved lower level particularly convex. Indeed, our methods may extends its applications to more problems with convexitor or reparametrization methods.
>
> 2.	(9) shows the theoretical influence of the discarding process to the learning accuracy, and the experiments of SAM [1] shows different loss of this discarding.
> [1] Foret et al., 2021. Sharpness-Aware Minimization for Efficiently Improving Generalization.
> 3.	(15) is the iterations of the algorithm(we move it to the appendix), is indeed a gradient desent iteration of $L^{b_k}$. This is based on the reformulation by the Moreau envelope and the penalty method in (14) for BLPPs. This algorithm updates the upper level variable together with de lower level--single loop—meaning that it do not need to solve the inner problem first and thus ensures the numerical efficiency. Besides, it take the intrinsic Hessian informations of L into considerations and thus balances the learning efficiency and numerical costs to solve the traditional SAM difficulties.
> 4.	We have add more explinations and backgrounds of some parts of this paper. Thanks for the reviewer’s valuable suggestions.
> 5.	Yes, we have mentioned potential techniques to extends our methods in Section 4. Thanks again for the valueable suggestions. Following with SAM-AT(ICML 24), we perform our algorithm for the adversarial training on the semantic segmentation task. We construct the DeeplabV3 with Resnet50 as backbone under Standford background dataset.  Our method outperforms SGD and AT across all settings.
>
>
> | Method         | Natural | $L_\infty$-AT | $L_2$-AT |
> |----------------|---------|-----------------|------------|
> | SGD            | 64.0    | 57.3            | 57.0       |
> | AT ($\epsilon=4/255$) | 55.3    | 54.9            | 55.1       |
> | Ours ($r=0.02$)   | **64.7** | **57.6**        | **57.2**   |

---

> > ### Comment · Reviewer_MepV · 2025-08-02
> >
> > Thank you for your response. I will make my final rating after discussing with other reviewers and AC.

---

> > > ### Comment · Reviewer_MepV · 2025-08-07
> > >
> > > I have reviewed the comments from the other reviewers, and I believe the author's rebuttal has addressed their concerns. I have no further questions or concerns. I will maintain my rating.

---

### Official Review · Reviewer_etJk · 2025-06-30

**Clarity:** 2
**Significance:** 2
**Originality:** 2
**Rating:** 4
**Confidence:** 3

**Summary:**

The paper proposes a bi-level optimization technique as an alternative to the usual min-max optimization algorithms. The authors study the convergence of the proposed algorithm and perform numerical experiments.

**Questions:**

Can we have more empirical evidence of the performance of the method? Also, how does the proposed method compare to, say, training a standard GAN in terms of computational cost?

**Ethical Concerns:**

["NO or VERY MINOR ethics concerns only"]

**Final Justification:**

After reviewing the author's rebuttal, and the additional experimental evidence they provided, I am raising my score to a 4.

**Limitations:**

The results in the paper are limited in significance. The claims of extensive experiments and improvements over existing methods are not justified properly.

**Quality:**

3

**Strengths And Weaknesses:**

Strengths:

- The problem is interesting and well motivated.
- The paper has both theoretical convergence results, and numerical experiments.


Weaknesses:

- The paper claims that the proposed method is better than the existing ones, but this claim is not supported by the experiments. The experiments on GANs are only done on the usual mixture of 8 Gaussians on a ring. It is known that the vanilla GAN can learn this distribution perfectly given the right hyperparameters. Moreover, it is very surprising for a  WGAN to have such poor performance on this dataset. This makes it hard to trust the results of Figure 2, since it is very easy to train a WGAN to learn this distribution. In the appendix, it is stated that the GANs were trained on 500 datapoints, which is extremely low for this dataset and neural networks.

- With this, it is not clear what improvements the proposed algorithm brings compared to existing min-max optimization techniques.

- Also, the use of the FID score on synthetic 2d datasets is not really meaningful, since FID is by definition tied to image datasets.

- I think the writing of Section 4 could be improved, it is a bit dense and hard to follow.

- Minor: typo on line 61 in the word "Learning"

---

> ### Author Rebuttal · Authors · 2025-07-31
>
> We thank the reviewers for their thoughtful feedback and careful evaluation. This work introduces a bilevel optimization framework that not only provides new theoretical insights beyond existing methods but also achieves strong practical performance and scalability across diverse domains. To address potential concerns regarding the theory–practice gap, we rigorously validate our framework through extensive large-scale experiments, including new GAN benchmarks against nine classical and state-of-the-art approaches, and demonstrate its extensibility to broader tasks such as adversarial learning. We believe these contributions establish both the novelty and practical relevance of our approach, and we respectfully request the reviewers to reconsider their evaluation in light of these results.
>
>  **New experiments of GAN:** To validate our method’s effectiveness in high-dimensional and complex settings, we conducted two high-dimensional experiments of GAN, comparing with nine typical GAN methods and will replace the section of GAN with these new results in the revised paper.
>
> Specially, we first conducted comparison with Stacked MNIST, a challenging dataset with 1000 modes. Obviously, Our method achieves full mode coverage (1000 modes) and the lowest $D_{KL}$ (0.09)，outperforming typical schemes (DCGAN, WGAN) and the latest baseline (R3GAN, NeurIPS 24), where our method is only trained with 1k steps.
>
> | Methods  | DCGAN | VEEGAN | WGAN | PacGAN | R3GAN | Ours  |
> |----------|-------|--------|------|--------|----------|-------|
> | Modes    | 99    | 150    | 959  | 992    | 1000     | **1000** |
> | $D_{KL}$ | 3.40  | 2.95   | 0.73 | 0.28   | 0.12     | **0.08** |
>
> We tested our method on CIFAR-10 using the same architecture as R3GAN (NeurIPS 2024, 21M parameters) with 50,000 training samples. FID results illustrate the performance of our schemes compared to state-of-the-art GANs.
> | Model      | Publication  | Parameters | FID   |
> |------------|--------------|------------|-------|
> | BigGAN     | ICLR 2019    | 82M        | 14.73 |
> | TransGAN   | NeurIPS 2021 | 10M        | 9.26  |
> | ViTGAN     | ICLR 2022    | 86M        | 6.66  |
> | StyleGAN3  | NeurIPS 2021 | 40M        | 10.83 |
> | R3GAN      | NeurIPS 2024 | 21M        | 6.21  |
> | Ours       | -            | 21M        | **5.89** |
>
> Convergence analysis over training steps also shows our method’s stability. In summary, our method achieves a 5% FID improvement over R3GAN at 2.4K steps and converges more stably, demonstrating superior performance and robustness in high-dimensional settings.
> | Step | 200   | 400   | 800  | 1000 | 1400 | 1800 | 2000 | 2200 | 2400 |
> |------|-------|-------|------|------|------|------|------|------|------|
> | Base | 33.09 | 16.98 | 8.75 | 7.50 | 6.92 | 6.55 | 6.21 | 6.26 | 6.37 |
> | Ours | 34.47 | 18.15 | 9.32 | 7.92 | 6.94 | 6.09 | 5.98 | 6.02 | 5.89|
>
> Furthermore, please also note that the experimental details of 2D mixture about GAN follows with BVFSM (TPAMI 23), where WGAN also cannot displays the desired performance.
>
> **Computation efficiency of GAN:** We argue that standard GANs rely on simple alternating optimization, lacking robust convergence theory and often requiring engineering tricks (e.g., gradient penalties) to avoid mode collapse.  Unlike GANs, our single-loop algorithm is supported by rigorous convergence analysis with first-order computation, achieving faster and more stable convergence. This is validated by a 16.8% efficiency improvement over BAMM (single-loop BLO scheme) and  more stable FID reduction on CIFAR-10, compared with the latest GAN scheme R3GAN .
>
> **Theoretical Foundations of Our Framework:**
> The technique in this paper is mainly based on the prior BLPP techniques and a reformulation of the min-max problem. Specifically, we build a stocstic version of the single loop algorithm and its convergence result which shows better numerical efficiency. And for the BLPP reformulation, as we maintain the intrinsic informations of classical min-max formulations (see the Example of SAM in Section 3), our method will conduct better effects then classical formulations without introducing too much numerical costs theoretically.
>
> **Revisions to Section 4 (Theoretical Part):** We have revised and reorganized several paragraphs to improve the clarity and quality of the writing.
>
> **Typos:** Thanks for pointing out these typos. We will check the paper carefully to revise all typos and mistakes.

---

> > ### Comment · Reviewer_etJk · 2025-08-01
> >
> > Thank you for your response and for providing additional experimental evidence. I also looked at your response to other reviewers and I am raising my score.

---

### Official Review · Reviewer_fVRv · 2025-07-02

**Clarity:** 3
**Significance:** 3
**Originality:** 4
**Rating:** 5
**Confidence:** 4

**Summary:**

This paper presents a bilevel reformulation of a broad class of non-convex min-max optimization problems by leveraging the Moreau envelope and efficient bilevel solvers. The goal is to transform unstable min-max problems into structured bilevel formulations that can be optimized more reliably. The authors argue that many min-max settings, such as those found in adversarial learning and sharpness-aware minimization (SAM) training, suffer from instability and convergence difficulties. Their proposed framework smooths the inner maximization, enabling the problem to be reformulated as two min-max problems, thereby facilitating faster optimization and providing theoretical convergence guarantees under appropriate conditions.

**Questions:**

Can the proposed bilevel reformulation be systematically applied to a broader class of min-max problems beyond the two case studies, and if so, what are the key structural requirements for reframing them as bi-level?

How does the proposed approach compare with those in Vlatakis et al. (2023)?

What challenges or limitations arise when scaling the bilevel reformulation to high-dimensional settings (e.g., deep generative models for vision tasks)?

**Ethical Concerns:**

["NO or VERY MINOR ethics concerns only"]

**Final Justification:**

During the rebuttal the authors clarified and addressed all of my concerns. Consequently, I have increased my already positive evaluation from 4 to 5.

**Limitations:**

Limitations are not explicitly discussed either in the main paper or in the checklist.

**Quality:**

3

**Strengths And Weaknesses:**

The paper is well-written and elegant in its mathematical presentation. It communicates complex optimization insights in an accessible and often tutorial style, which is valuable for readers looking to understand the deeper structure of bilevel optimization. In this sense, the paper is intellectually stimulating and has educational value. While some definitions and derivations (e.g, derivation of Table 1 or the definition of the Moreau envelope) could be made more explicit (or more directly linked to earlier assumptions, overall the exposition is smooth. The choice to apply the method to two different problems helps demonstrate the potential generality of the framework, even though, as discussed below, the scope of the experiments remains limited.

The main limitations I see relate to the scope of the experimental validation and the clarity of the practical motivation and applicability of the proposed framework.

First, the empirical section on generative adversarial learning (Section 5.1) is limited to a two-dimensional toy example, despite being presented under the heading of "Real-world applications". A low-dimensional GAN example is not sufficient to demonstrate practical utility or scalability. Generative models, especially adversarial ones, typically operate in high-dimensional spaces (e.g., image generation), where both the generator and discriminator are deep networks with thousands or millions of parameters. It thus remains unclear whether the proposed bilevel reformulation can offer tangible advantages (computational or robustness-related) in these more practical settings. How does the approach scale? Is the inner problem still tractable when dealing with large-scale models and complex data distributions? Can the framework, in practice, compete with existing heuristics used in adversarial learning in terms of speed and quality of the results?

Second, while the goal of the paper is to reframe min-max problems as bilevel problems, it is not sufficiently clear **why** this is beneficial beyond the ability to apply specific solvers. The authors mention robustness benefits (e.g., line 84), but they do not demonstrate whether these benefits stem from the reformulation itself or the improved optimization dynamics afforded by better-behaved algorithms. Is the observed robustness in classification simply a result of smoother convergence, or is there a deeper benefit to recasting the problem as bilevel in structure beyond its theoretical framework?

Moreover, the paper lacks a general guideline or procedure for reformulating generic min-max problems into bilevel ones. While the two case studies help illustrate the method, they do not offer a systematic view. Readers unfamiliar with the particular structure of the loss functions involved may struggle to apply this technique to other use cases. It would strengthen the work significantly to provide a more general prescription for when and how this reformulation applies, along with its assumptions and limitations. At the very least, this limitation should be explicitly acknowledged in the paper.

Lastly, the paper does not engage meaningfully with related work that attempts to solve min-max problems via alternative structure-aware methods. In particular, the recent work by Vlatakis et al. (2023) proposes gradient descent-ascent methods tailored for min-max problems with hidden structure. Given the similarity in motivation and objectives, a more thorough discussion comparing these two perspectives would be highly valuable. At the very least, the paper should provide a conceptual framework for its bilevel reformulation of existing methods. It would also be important to clarify whether bilevel approaches offer distinct computational or theoretical advantages over the work by Vlatakis et al. (2023), and under what specific conditions these advantages hold.

[Vlatakis et al.] Solving min-max optimization with hidden structure via gradient descent ascent. NeurIPS 2023.

*Minor Observations:* The derivation of Table 1 is not comprehensively explained, and the Moreau envelope definition appears somewhat unexpectedly, without strong contextual framing.


**Conclusion.** The current experimental evidence is not sufficient to validate its utility in real-world, high-dimensional applications, particularly in the context of generative modeling. The paper would benefit from a more convincing empirical demonstration, stronger motivation for the reformulation in practical terms, and a more precise comparison with other recent advances in min-max optimization.

---

> ### Author Rebuttal · Authors · 2025-07-31
>
> **Comparison under high-dimensional applications**:
> Firstly, to address the reviewer’s concern about high-dimensional applicability, we conducted new experiments on CIFAR-10 with our method using the same network as R3GAN. Our method achieves a 5% FID improvement over R3GAN, only training with 2.4k steps, demonstrating competitive performance in image generation tasks.
> | Model      | Publication  | Parameters | FID   |
> |-------|-------|-------|-------|
> | BigGAN     | ICLR 2019    | 82M        | 14.73 |
> | TransGAN   | NeurIPS 2021 | 10M        | 9.26  |
> | ViTGAN     | ICLR 2022    | 86M        | 6.66  |
> | StyleGAN3  | NeurIPS 2021 | 40M        | 10.83 |
> | R3GAN      | NeurIPS 2024 | 21M        | 6.21  |
> | Ours       | -            | 21M        | **5.89** |
>
> Furthermore, our experiments on SAM (training with diverse typical bakcbones) and large-scale numerical cases also demonstrate the scalability. Comparing existing heuristic and bi-level optimization method, our method realize the consistent improvement. We are committed to revising the manuscript to incorporate these clarifications and results.
>
> **Theoretical Benefit of Bilevel Reformulation:**
> The reviewer rightly questions whether robustness stems from the reformulation itself or algorithmic improvements. We emphasize that the advantages:
>
> •	Intrinsic Stability: By decoupling the min-max dynamics via Moreau smoothing, our reformulation inherently avoids cyclic gradients (e.g., Figure 4 in Vlatakis et al.), even with vanilla solvers (Appendix C.3 validates this with fixed-step GDA).
>
> •	Accuracy Preservation: Unlike heuristic simplifications (e.g., SAM’s "discarding step"), our bilevel framework preserves essential problem structure, enabling theoretical guarantees (Theorem 4.1).
>
> **General Guidelines for Reformulation:** General Guidelines for Reformulation
> To clarify applicability, we will add a step-by-step flowchart outlining:
> 1.	Problem Screening: Check if the classical method sacrifies learning accuracy for the numerical cost.
> 2.	Reformulation: Apply the bilevel reformulation, by reparametrization or convexitor if necessary.
> 3.	Algorithm: Apply the single-loop algorithm for the reformulation.
> This framework already covers SAM and GANs, and we explicitly discuss limitations:
>
> **Comparison to Vlatakis et al. (2023):**
> While both works address min-max instability, our approach differs fundamentally:
> | Aspect| Vlatakis et al.| Our Method|
> |------------|--------------|------------|
> | Assumptions | hidden convex-concave structure    | Smoothness |
> | Convergence Mechanism| Center-stable manifolds | Bilevel constraints (Theorem 3)|
> | Aim | Convergence of hidden convex-concave games    | Balancing learning efficiency and numerical cost |
>
> Our method bypasses cyclic attractors (Vlatakis’ Fig. 4) by bilevel methods.
>
> **Future Generalization.** For various types of min-max problems, modern methods have been proposed (e.g., [andriushchenko2020understanding, andriushchenko2022towards, li2020towards, zhang2022revisiting, vlatakis2021solving]). For instance, vlatakis2021solving investigates min-max problems of the form$\min_{\theta}\max_{\phi}L(F(\theta),G(\phi))$ in the convex-concave setting. Our method can be applied to some of these problems, particularly when the inner concave subproblem $\max_{\phi}L(F(\theta),G(\phi)) $ is tractable, but solving the overall min-max problem requires sacrificing learning accuracy for computational efficiency. Some of these problems suffer from cyclic behavior of level sets (see Figure 4 in vlatakis2021solving). Unlike the approach in vlatakis2021solving, we address this issue through a bilevel reformulation, whose convergence is established in Theorem~3.
>
> In such cases, our bilevel framework provides a compelling alternative, offering a more favorable balance between learning accuracy and computational cost, as illustrated in the SAM and GAN applications.
>
> Furthermore, while not all min-max problems naturally satisfy the two features outlined earlier, many can be transformed into a suitable bilevel form through reparametrization techniques or the use of convexificators. This enables broader applicability of our approach, which can be extended via the following three-step process:
>
> - **Problem Screening**: Identify whether existing methods trade off learning accuracy for reduced numerical cost.
> - **Reformulation**: Apply a bilevel reformulation, using reparametrization or convexification techniques if needed, to ensure the lower-level problem is tractable.
> - **Algorithm Deployment**: Solve the reformulated problem using the proposed single-loop bilevel algorithm.
> This process broadens the applicability of our framework beyond SAM-type problems. Investigating such extensions remains an important direction for future research.
>
> **Limitations.** Beyond the SAM and GAN problems discussed in this paper, our method is applicable to a broader class of min-max problems that possess the following characteristics:
>
> 1). Classical methods for min-max problems often trade learning accuracy for numerical efficiency, as the inner-level optimization is typically difficult to solve. In such cases, introducing a bilevel reformulation allows us to better balance learning accuracy and computational cost.
>
> 2). The original min-max problem can be reformulated as a bilevel problem in which the lower-level problem is easily solvable, without requiring any convexity assumptions on the objective function $L$. In such settings, the bilevel reformulation does not introduce significant additional complexity.
>
> Given these two features, the bilevel approach offers distinct advantages. On the one hand, it preserves more intrinsic and essential information from the original problem than standard simplifications (e.g., the “discarding step” used in SAM), thereby enhancing learning accuracy. On the other hand, when the lower-level problem  is easily solvable, efficient bilevel optimization algorithms can be employed, resulting in a computational cost that remains manageable.
>
> Nevertheless, in high-dimensional learning tasks, our method may encounter challenges in maintaining an acceptable computational cost while ensuring learning accuracy. This difficulty, however, is inherent to high-dimensional settings: higher-dimensional data typically carry more information, which inevitably leads to increased computational demands. In such cases, the trade-off between accuracy and efficiency becomes more pronounced, further emphasizing the value of a principled framework for balancing the two.
>
> **Table 1 Analysis:** Table 1 presents three key metrics for analyzing loss landscape geometry. The ascent-direction loss, corresponding to the minimum eigenvalue (λₘᵢₙ) of the Hessian matrix H_L(ω), captures the local curvature along the most favorable ascent direction. The average-direction loss, quantified by the trace of the Hessian (Tr(H_L(ω))), characterizes the overall sharpness of the loss surface. The table clearly demonstrates that SAM's effectiveness in sharpness minimization stems from its utilization of second-order information through the Hessian matrix. Notably, the largest eigenvalue (λ₁(H_L(ω))) serves as a direct measure of loss function sharpness, thereby elucidating SAM's mechanism for improving generalization via sharpness reduction. For detailed mathematical derivations of these relationships, we refer readers to Theorems E.1-E.3 in [Wen et al., 2023].
>
> **Moreau Envelope:** To address the nonsmooth nature of the value function, we employ the Moreau envelope technique to construct a smooth approximation. This approach not only enhances mathematical tractability but also facilitates algorithmic development. Geometrically, the Moreau envelope generates a family of smooth surfaces that progressively approximate the original value function's graph. We will revise this part for the clear description.

---

> > ### Comment · Reviewer_fVRv · 2025-08-02
> >
> > Thank you for the detailed rebuttal. From my side, I kindly invite the authors to address two remaining points.
> >
> > First, please provide more details about the experimental configuration for the new CIFAR-10 experiments. Specifically, how much data was used, what training protocol and hyperparameter choices were followed (e.g., optimizer, learning rate, batch size, total training time) across all methods, and how stable the training dynamics were across runs. Additionally, it would be useful to know whether the bilevel approach required any significant tuning compared to the other baselines.
> >
> > Second, the argument that the bilevel formulation preserves problem structure and thus enables theoretical guarantees (e.g., Theorem 4.1) is compelling but currently underdeveloped. Could the authors elaborate on how this preservation translates into practical benefits? For example, in what scenarios would heuristic methods such as SAM’s simplifications lead to degraded performance or invalid solutions that the bilevel formulation avoids? Including a small illustrative example or a more direct comparison could help make this point more concrete.
> >
> > Overall, I would like to thank the authors again for their efforts and valuable work. I look forward to seeing their responses.

---

> > > ### Author Response · Authors · 2025-08-03
> > >
> > > We appreciate the reviewer’s valuable feedback.
> > >
> > > （1）To demonstrate our bilevel optimization framework’s effectiveness, we evaluated it on the standard CIFAR-10 benchmark using identical training protocols, batch sizes, learning rate, and hyperparameters as R3GAN (NeurIPS 2024). This highlights our framework’s flexibility, seamlessly integrating with existing methods, as evidenced in our GAN experiments.
> > >
> > > Convergence analysis reveals our method’s superior stability:
> > >
> > > | Step | 200   | 400   | 800  | 1000 | 1400 | 1800 | 2000 | 2200 | 2400 |
> > > |------|-------|-------|------|------|------|------|------|------|------|
> > > | R3GAN | 33.09 | 16.98 | 8.75 | 7.50 | 6.92 | 6.55 | 6.21 | 6.26 | 6.37 |
> > > | Ours  | 34.47 | 18.15 | 9.32 | 7.92 | 6.94 | **6.09** | **5.98** | **6.02** | **5.89** |
> > >
> > > Our method achieves a 5% FID improvement (5.89 vs. 6.37) with fewer oscillations, stabilizing by 1800 steps. While requiring more first-order gradient updates (7.53 hours vs. 4.72 hours for 2400 steps on an NVIDIA RTX 4090 GPU), our approach ensures robust mode coverage, unlike standard GANs, which rely on alternating optimization prone to mode collapse and dependent on heuristic tricks (e.g., gradient penalties). Our single-loop algorithm , supported by rigorous convergence analysis, uses first-order computations to achieve faster and more stable convergence.
> > >
> > > （2）To demonstrate the structural advantages of the bilevel reformulation over heuristic methods such as SAM, consider a simple anisotropic quadratic loss:
> > > $\[
> > > L(\omega) = \tfrac{1}{2} \omega_1^2 + 10 \omega_2^2, \omega \in \mathbb{R}^2
> > > \]$.
> > >
> > > This function has a flat direction along $\omega_1$ and a sharp one along $\omega_2$. In the SAM framework, the ascent direction $\delta$ is given by
> > > $\[
> > > \delta^{\text{SAM}} = \frac{\nabla L(\omega)}{\|\nabla L(\omega)\|}
> > > \]$,
> > >
> > > which tends to align with the steeper $\omega_2$ direction due to the larger curvature. The SAM update then evaluates $\nabla L(\omega + r \delta^{\text{SAM}})$ and proceeds accordingly. However, this heuristic update discards second-order structure, potentially leading to updates in sharp directions and bypassing flatter minima.
> > >
> > > In contrast, the bilevel formulation retains this structural information. The ascent direction is still $\delta^* = \arg\min_{\delta \in \mathcal{C}} -\delta^T \nabla L(\omega)$, but the outer optimization is performed on the perturbed input $L(\omega + r \delta^*)$, rather than its gradient. This perspective allows the optimizer to move toward flatter regions (e.g., along $\omega_1$), thereby improving generalization.
> > >
> > > This example demonstrates how the bilevel reformulation more accurately preserves curvature information in the original loss landscape, consequently yielding superior solutions. Extensive empirical results further confirm the consistent manifestation of this advantageous property across various experimental settings.

---

> > > > ### Comment · Reviewer_fVRv · 2025-08-05
> > > >
> > > > Thank you for your responses, I personally learned a lot from this discussion. I am raising my score.

---

### Note · Authors · 2025-08-15

We would like to sincerely thank all reviewers for their thoughtful and constructive feedback throughout the review process. The questions, suggestions have been extremely valuable in helping us improve the clarity, technical rigor, and presentation of our work. In particular, we have addressed all four reviewers’ comments in detail during the rebuttal and discussion phase, incorporating clarifications, additional examples, and revised explanations to strengthen both the theoretical and experimental components of the paper.

The reviewers’ input has not only helped us refine the arguments, such as providing clearer illustrative cases for the advantages of the bilevel formulation over heuristic simplifications, but has also led us to improve the organization and readability of the manuscript. We are grateful for these insights, which have significantly improved the quality of the work.

We would also like to thank the Area Chair for carefully moderating the discussion, as well as the Program Chairs and Organizing Committee for providing this platform that enables such meaningful exchange and learning. Regardless of the final decision, this review process has been very beneficial for us, and we deeply appreciate the opportunity to engage with the community in this way.

---

### Decision · Program_Chairs · 2025-09-17

**Decision:**

Accept (poster)

**Comment:**

(a) This work reformulates adversarial learning as a bilevel optimization problem, enabling more efficient and accurate training without added complexity. The authors propose a provably convergent single-loop stochastic algorithm that improves robustness and efficiency. Experiments show gains in both GAN generation quality and Sharpness-Aware Minimization (SAM) accuracy under noise, highlighting the practical benefits of the approach.

(b) This paper provides a theoretical contribution with rigorous convergence results for the bilevel reformulation. It is also well-written and explains well the structure of min-max problems. It has practical value: requires no added computational overhead while showing empirical benefits.

(c) Experiments are incomplete. GAN results restricted to toy datasets, so that scalability to high-dimensional, real-world tasks remains unclear.
The novelty is unclear. Some reviewers felt the approach builds on existing optimization techniques without introducing fundamentally new mechanisms. Some comparisons are missing. There is insufficient benchmarking against sota min-max methods. Although there is a theoretical contribution, it requires a convexity assumption, which is not available in general with deep neural networks.

(d) The theoretical results, conceptual clarity, and potential generality of the bilevel reformulation make this a valuable contribution. While the experimental validation is limited and novelty may be incremental, the framework provides new perspective on adversarial learning and optimization, with convergence results that strengthen its impact. Reviewers raised concerns about scalability and comparisons, but these were addressed during rebuttal, leading to improved scores.

(e) Reviewers initially highlighted concerns about limited experiments, lack of comparisons to recent methods, and unclear benefits beyond theory. Specific issues included reliance on toy GAN datasets, insufficient exploration of scalability, and ambiguity about novelty. In rebuttal, the authors clarified their framework, addressed methodological questions, and added experimental evidence, which reviewers found convincing. Reviewer fVRv raised their score from 4 top 5, etJk from 3 to 4, and KhNq from borderline reject to borderline accept, while MepV maintained a strong accept. The consensus after rebuttal converged toward acceptance, with recognition of theoretical soundness and practical promise despite limited empirical scope.